# Evaluating Graph Generative Models with Contrastively Learned Features

**Hamed Shirzad**
UBC
shirzad@cs.ubc.ca

**Kaveh Hassani**
Autodesk AI Lab
kaveh.hassani@autodesk.com

**Danica J. Sutherland**
UBC & Amii
dsuth@cs.ubc.ca

## Abstract

A wide range of models have been proposed for Graph Generative Models, necessitating effective methods to evaluate their quality. So far, most techniques use either traditional metrics based on subgraph counting, or the representations of randomly initialized Graph Neural Networks (GNNs). We propose using representations from contrastively trained GNNs, rather than random GNNs, and show this gives more reliable evaluation metrics. Neither traditional approaches nor GNN-based approaches dominate the other, however: we give examples of graphs that each approach is unable to distinguish. We demonstrate that Graph Substructure Networks (GSNs), which in a way combine both approaches, are better at distinguishing the distances between graph datasets. The code used for this project is available in: `https://github.com/hamed1375/Self-Supervised-Models-for-GGM-Evaluation`.

## 1 Introduction

Quantitative evaluation of generative models is challenging [45]; evaluating purely by visual inspection can introduce biases, particularly towards "precision" at cost of "recall" [37]. The traditional metric, likelihood, is also not only difficult to evaluate for implicit generative models on complex, highly structured datasets, but can also be a poor fit to users' goals with generative modeling [37, 45]. Many quantitative proxy measures for the discrepancy between generator and real distributions have thus been used in recent work on generative modeling of images, some of the most important including the Inception Score (IS) [38], Fréchet Inception Distance (FID) [20], Precision/Recall (PR) [37], and Density/Coverage [32]. FID and PR measures in particular have been shown to correlate with human judgments in some practical settings. All of these measures use representations extracted from CNNs pretrained on ImageNet classification [36]. Nevertheless, it is recently shown that self-supervised representations archives more reasonable ranking in terms of FID/Precision/Recall, while the ranking with ImageNet-pretrained embeddings often can be misleading [30].

These problems are exacerbated for the evaluation of generative models of graph data. Graphs are often used to represent concepts in a broad range of specialized domains, including various fields of engineering, bioinformatics, and so on, so that it is more difficult or perhaps impossible to find "generally good" features like those available for natural images via ImageNet models. Many methods for evaluating graph generative models are thus based on measuring discrepancies between statistics of generic low-level graph features, including local measurements like degree distributions, clustering coefficient distributions, and four-node orbit counts [24, 52], or simple global properties such as spectra [25]. Differences between distributions of these features are generally measured via the maximum mean discrepancy (MMD) [15] or total variation (TV) distance.

More recent work [46] proposes instead extracting features via randomly initialized Graph Isomorphism Networks (GINs) [51]. These features provide a representation for each graph, so that metrics like FID, Precision/Recall, and Density/Coverage can be estimated using these representations. For

36th Conference on Neural Information Processing Systems (NeurIPS 2022).

datasets with natural labels, we can alternatively find representations based on training a classifier; such features, however, are optimized for a very different task.

One significant challenge of graph data is that it is computationally difficult to even tell whether two graphs are the same (are isomorphic to one another). GINs are known to be as powerful at detecting isomorphism as the standard Weisfeiler-Lehman (WL) test [51], but it is not known whether *random* GIN representations have this ability. Experimental evidence [51] shows that trained GIN networks are as powerful as the WL test, but have substantially worse power at random initialization.

Unlike for distributions of natural images, however, the relevant graph features and particularly the semantics of node or edge features varies widely between datasets. On molecular graphs, adding any single edge will almost always violate physical constraints about stable atomic bonds; for house layouts, the feasibility of adding an edge depends significantly on the rest of the layout and the type of rooms (few front doors open into a bathroom, and direct connections between rooms on the first and third floors are unlikely); Traditional metrics and random graph neural networks are both unlikely to be able to capture these complex inter-dependencies functioning very differently across domains, and it similarly seems quite unlikely that pretraining on some "graph ImageNet" would be able to find useful generic graph features.

We therefore propose to train graph encoders on the same data used to train the generative model, using self-supervised contrastive learning to find meaningful feature extractors [18, 19, 34, 41, 44, 48, 53–56], which we then use to compare the generated graphs to a test set. The set of perturbations introduced in contrastive learning teaches the model which kinds of graphs should be considered similar to one another. In this work we use types of perturbations traditionally used for training contrastive learning methods on graphs, but point out that future work focusing on domain-specific modeling can directly incorporate knowledge about which graphs should be considered similar by choosing different perturbation sets.

Inspired by the theoretical results for contrastive learning of [50], we also propose two variations to our representation learning procedures. This upper bound shows that contrastively learned representations work well for downstream tasks as long as the probability of data points yielding overlapping augmentations is relatively large for *within-class* dataset pairs and relatively small for *cross-class* pairs. Edit distances between graphs on typical training sets are large, however; we propose to use subgraphs ("crops") in our set of data augmentations for contrastive learning, which is more likely to yield overlaps. We also suggest enforcing a layer-wise Lipschitz constraint on feature extractors, which encourages similar graphs to have similar learned representations. We show experimentally that both changes improve learning.

With all of these improvements, we further ask: can we theoretically guarantee that our learned GNN representations outperform traditional local metrics? We prove that we cannot: we give examples of graphs easily distinguishable by local metrics that first-order GNNs cannot distinguish. Yet the converse is also true: we show graphs easily distinguishable by GNNs that appear equivalent to local metrics. We thus propose to use models based on Graph Substructure Networks (GSNs) [6], (using node degrees, and node clustering coefficients), and as a result, explicitly incorporating local metrics into our models and surpassing the power of the WL test, and yield further improvements.

## 2 Related Work

**Graph generative models.**   Our work is not particular to any type of graph generative model, as it focuses on simply evaluating samples; nonetheless, it is worth briefly reviewing some methods. Sequential generation models, such as GraphRNN [52] and GRAN [25], generate nodes and edges in an auto-regressive manner. These models are efficient, but can struggle to capture global properties of the graphs, and require a predefined ordering. One-shot models such as MolGAN [10], GraphDeconv [13], GraphVAE [23], and GraphNVP [27], on the other hand, generate all nodes and edges in one shot; they can model long-range dependencies, but generally are not efficient and cannot scale to large graphs. For a detailed overview of graph generative models, see [16].

**Evaluating graph generative models.**   Graph generative models can be even more challenging to evaluate than visual or textual models, because it is generally more difficult for humans to judge the quality of a generated graph. The classic measure of the quality of a generative model, likelihood, also has significant issues with graphs: in addition to the kinds of issues that appear in generative

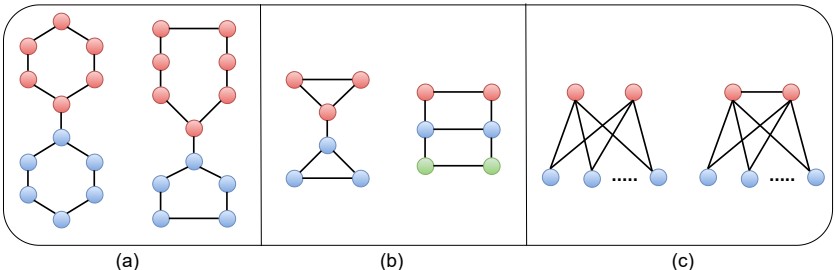

**Figure 1:** (a) An example of two graphs that can be differentiated by GNNs but not local metrics. Both graphs have same degree distribution, clustering coefficient and 4 node orbits. (b) An example of two graphs that can be differentiated by local metrics (clustering coefficient and smallest cycle) but not GNNs. (c) Adding a single edge can drastically change cluster coefficient and 4-node orbits.

models of images [45], the likelihood is particularly hard to evaluate on graphs where even checking equality is quite difficult [33]. The most common method is to compute the Maximum Mean Discrepancy (MMD) [15] between distributions of local graph statistics, including node degrees, cluster coefficients, and counts of orbits up to a particular size [25, 52]. Global statistics, such as eigenvalues of the normalized graph Laplacian, are also used [52]. These metrics, however, focus only on low-level structure of the graph, and ignore any features that might be present on nodes or edges [46]. Choosing an appropriate kernel is also very important to consistency of these metrics [33]. These types of graph statistics might also encourage models to overfit to the training data, rather than truly learning the target distribution [40].

For unlabeled image-based generative models, most work focuses on metrics including the Inception Score (IS) [38], Fréchet Inception Distance (FID) [20], Precision/Recall (PR) [37], and Density/Coverage [32], all of which compare distributions in a fixed latent space (typically activations of a late layer in an InceptionV3 [43] model trained on ImageNet). These methods are rarely adopted in graph generative models, due to challenges with "general-purpose" graph models discussed in Section 1. Thompson et al. [46] thus used random (untrained) graph encoders in these metrics. We discuss these methods in more detail in the appendix. Our work, inspired by Morozov et al.'s similar proposal in image domains [31], explores the use of self-supervised contrastive training to find representations that work better than random initializations.

**Graph Contrastive Learning.** A popular method for self-supervised learning, contrastive learning generally aims to find a representation roughly invariant to various operations (e.g. for images, taking random crops, horizontal flipping, shifting colors) but able to identify different source data points. Ideally, such a representation will be useful for downstream tasks not known when learning the representation. In graph settings, learned representations may be at the node, edge, or graph levels.

Early works adopting contrastive learning to encode graphs used DeepInfoMax [21] loss to enforce consistency between local (node) and global (graph) representations (e.g., DGI [48] and InfoGraph [41]). These methods did not use any specific augmentations and simply used distinct graphs as negative examples. Following works begin utilizing graph augmentations to further improve the learned representations. For example, MVGRL [18] uses graph diffusion and sub-graph inducing as two types of augmentations, where as GCC [34] only uses multi-hop ego network inducing. More recent works use four types of graph augmentations including: feature masking, node dropping, edge perturbation, and sub-graph inducing [19, 44, 53–56]. Among them, a few works rely on trial-and-error or heuristics to choose those augmentations for a given dataset (e.g., GraphCL [54], GRACE [55], and GCA [56]), whereas others introduce a policy network to learn to sample augmentations and compute their parameters end-to-end along with graph representations (e.g., JOAO [53] and LG2AR [19]).

## 3 GNNs Versus Local Metrics

Different methods for understanding graphs can "understand" the difference between graphs in very different ways: a particular change to a graph might barely affect some features, while drastically changing others. One extreme case is when a given metric cannot detect that two distinct (non-

isomorphic) graphs are different. Since graph isomorphism is a computationally difficult problem, we expect that all efficiently computable graph representations "collapse" some pairs of input graphs.[1] It is conceivable, however, that one method could be strictly more powerful than another. For instance, since recent GNN models have overcome traditional models based on local metric representations in a variety of problems [23, 47, 51], is it the case that GNNs are strictly more powerful than local metrics?

We show, constructively, that the answer is no: there are indeed graphs which GNNs can distinguish and local metrics cannot, but there are also graphs which local metrics can distinguish but first-order GNNs cannot. Figure 1(a) shows a pair of graphs with the same degree distribution, clustering coefficient, and four-node orbits, which can nonetheless be distinguished by GNNs (proof in Appendix B). On the other hand, the graphs in Fig. 1(b) have different clustering coefficient and smallest cycle, but first-order GNNs cannot tell them apart. Thus, neither method strictly outperforms the other on all problems, and so there are theoretical generative models which perfectly match in GNN-based representations but differ in local metrics, and vice versa. This motivates our addition of local features to our graph representation models (Section 4).

It is much easier to incorporated such hard-coded structures into GNNs than it would be to add learning to feature metrics; in particular, counting higher-order local patterns quickly becomes prohibitively expensive, with a super-exponential time complexity. GNNs can also easily handle node and/or edge features on the underlying graphs, which is far more difficult to add to local metrics.

Another quality we would like in our graph representations, in addition to the ability to distinguish distinct graphs, is some form of stability: if a distribution of graphs only changes slightly, we would like our evaluation methods to give "partial credit" as opposed to a distribution where all graphs are dramatically different. (This is closely related to issues in training and evaluating image-based generative models [1–3, 45].) As previously discussed, the notion of "similar graphs" is very domain-dependent, but traditional local metrics can be highly sensitive to changes like adding a single edge: Fig. 1(c) shows an example where two graphs differing only by a single edge can have drastically different statistics. By learning GNN representations, we can have some control over these types of smoothness properties; we exploit this explicitly in our methodology in Section 4.

## 4 Self-supervised Training of Graph Representations for Evaluation

Suppose we have two sets of graphs $\mathbb{G}^{train} = \{G_1, \dots G_S\}$ and $\mathbb{G}^{test} = \{G_1, \dots G_M\}$, each sampled from the same data distribution $p(G)$. Also suppose that we have access to an unconditional graph generative model $g_\phi(.)$, which is trained on $\mathbb{G}^{train}$ to learn the distribution of the observed set of graphs. We sample a set of generated graphs $\mathbb{G}^{gen} = \{G_1, \dots G_N\} \sim p_{g_\phi}(G)$ from this model. In order to evaluate the quality of the sampled graphs (i.e., to decide whether the model $g_\phi(.)$ has successfully recovered the underlying distribution $p(G)$), we can define a measure of divergence $\mathcal{D}(\mathbb{G}^{test}, \mathbb{G}^{gen})$ to quantify the discrepancy between distributions of the real and generated graphs. One robust way to achieve this is to define the metric on latent vector representation spaces, and expect representations of graphs rather than the original objects. Thus, to use these metrics, we need to train a shared encoder $f_\theta(.)$ and then compute the discrepancy as $\mathcal{D}(f_\theta(\mathbb{G}^{test}), f_\theta(\mathbb{G}^{gen}))$. There are a few such metrics well-studied in visual domains that can differentiate the fidelity and diversity of the model, and which we can adopt in graph domains.

For evaluating image generative models, due to similar feature space across image datasets and also availability of large-scale data, the trunk of a model trained over ImageNet with explicit supervisory signals is usually chosen as the encoder. However, it is not straightforward to adopt the same trick to graph-structured data: there are no ImageNet-scale graph datasets available, and more importantly, the semantics of graphs and their features vary wildly across commonly used graph datasets, far more than occurs across distributions of natural images. For instance, even datasets of molecular graphs may use different feature sets to represent the molecules [17], in addition to the many cross-domain challenges discussed in Section 1. Thus, it is not feasible to imagine a single "universal" graph representation; we would like a general-purpose method for finding representations useful for a new graph dataset.

---

[1]Alternatively, an efficient graph representation might be able to distinguish non-isomorphic graphs, if it also sometimes distinguishes isomorphic graphs. We do not consider such representations in this paper.

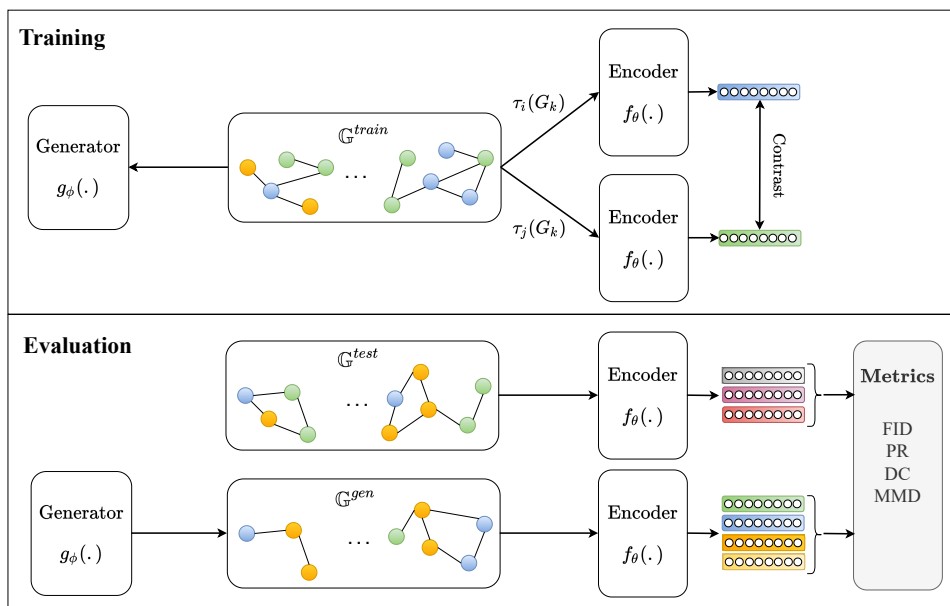

**Figure 2:** During the **training** phase, we use the same training data ($\mathbb{G}^{train}$) to train both the generator and the encoder networks. The encoder is trained using a contrastive loss where two augmentation $\tau_i$ and $\tau_j$ are randomly sampled from a set of rational augmentations $\mathcal{T}$ to construct two views of a sampled graph $G_k$. During **evaluation** phase, we sample the generator to form a generated set of graphs $\mathbb{G}^{gen}$ and feed it along with a held-out set of real graphs $\mathbb{G}^{test}$ to the encoder to compute the graph representations. The representations are then used to compute robust metrics to quantify the discrepency between real and generated graphs.

**Training with Graph Contrastive Learning**   To find expressive representation of real and generated graphs, we train the encoder using a contrastive objective. Assuming a set of rational augmentations $\mathcal{T}$ over $\mathbb{G}$ where each augmentation $\tau_i \in \mathcal{T}$ is defined as a function over graph $G_k$ that generates an identity-preserving view of the graph: $G_k^+ = \tau_i(G_k)$, a contrastive framework with negative sampling strategy uses $\mathcal{T}$ to draw positive samples from the joint distribution $p\left(\tau_i(G_k), \tau_j(G_k)\right)$ in order to maximize the agreement between different views of the same graph $G_k$ and to draw negative samples from the product of marginals $p\left(\tau_i(G_k)\right) \times p\left(\tau_j(G_{k'})\right)$ to minimize it for views from two distinct graphs $G_k$ and $G_{k'}, k \neq k'$.

As mentioned previously, we use a GIN architecture for our feature extractor. Other than the training process, the rest of our evaluation pipeline is similar to that of Thompson et al. [46], who use random GIN weights. We consider two methods for training our GIN's parameters: GraphCL [54] and InfoGraph [42]. GraphCL randomly samples augmentation from four possible augmentaiotns including *node dropping*, *edge perturbation*, *attribute masking*, and *sub-graph inducing* based on a random walk to which we would like the representation to be roughly invariant. GraphCL uses normalized temperature-scaled cross-entropy (NT-Xent) objective [7] to maximize the agreement between positive graph-level representations. InfoGraph works differently: it contrasts the graph-level representation with the representations of individual nodes, which encode neighborhood structure information. InfoGraph uses DeepInfoMax [21] objective to maximize the mutual information between graph-level and node-level representations of each individual graph.

**Using Local Subgraph Counts as Input Features for GINs**   To build on the insight of Section 3, we also consider various methods for adding information about the local graph structure as node features, similarly to Graph Substructure Networks [6]. The simplest such method is to add the degree of a node as an explicit feature for the GIN to consider. We do this, but also add, by concatenation on node features, higher-order local information as well: three-node and four-node clustering features for each node. Four node clustering coefficient is calculated as:

$$C_4(v) = \frac{\sum_{(u,w)\in\mathcal{N}(v)} q_v(u,w)}{\sum_{(u,w)\in\mathcal{N}(v)} \left[deg(u) + deg(w) - q_v(u,w) - 2\mathbb{I}(u \in \mathcal{N}(w))\right]} \tag{1}$$

where $\mathcal{N}(v)$ denotes immediate neighbors of node $v$, $deg(v)$ denotes the degree of node $v$, and $q_v(u, w)$ is the number of common neighbors of $u$ and $w$, not counting $v$. Aggregating these features across the whole graph would give information on distribution of 4 node orbits of the graph, but this provides more localized information across the graph that is nonetheless difficult or impossible for a GIN to examine otherwise [8].

**Choice of Augmentations**    Wang et al. [50], building off work of Wang and Isola [49], study contrastive learning in general settings (with an eye towards vision applications), and provide an intriguing bound for the performance of downstream classifiers. To explain it, consider the "augmentation graph" of the training samples: if $G_i$ is the $i$th training example and $G_i^+$ is a random augmentation of that example, we connect the nodes $G_i$ and $G_j$ in the augmentation graph if there are feasible augmentations $G_i^+$ and $G_j^+$ such that $G_i^+ = G_j^+$.[2] Wang et al. argue that downstream linear classifiers are likely to succeed if this augmentation graph has stronger intra-class connectivity than it does cross-class connectivity, proving a tight connection between the two under a particular setting with strong assumptions about this and related aspects of the setup.

Given the connection between the distribution metrics discussed in Appendix A.2 and classifier performance [e.g. 26, Section 4], if we accept the argument of Wang et al. [50], evaluation metrics based on a contrastively trained graph representation will give poor values (good classifier performance) when generated samples are not well-connected to real samples in the augmentation graph, and vice versa. If we choose augmentations appropriately, this is sensible behavior.

**Enforcing Lipschitz Layers in Representation Networks**    The prior line of reasoning also suggests that we should choose augmentations that are capable of making real graphs look like one another. Edit distances between graphs, however, are typically large on the datasets we consider, and so augmentations based on adding or deleting individual nodes and/or edges will struggle to do this. The same is true for many of the augmentations used on images, except – as Wang et al. [50] note – for crop-type augmentations, where e.g. two different car pictures might become quite similar if we crop to just a wheel. On graphs, an analogous operation is subgraph sampling, which we include in our GraphCL setup; InfoGraph already naturally looks at subgraph features as a core component.

Taking this line of reasoning as well as the general motivations of contrastive learning further, it is also natural to think that if we can inherently enforce "similar graphs" to have similar representations, this could improve the process of contrastive learning: we would save on needing to train the model to learn these similarities, and it could help decrease the classifier performance for good generative models whose output graphs are legitimately near the distribution of target graphs.

A line of work on GANs in visual settings [1–3, 28, 29, 35] has made clear the importance of this type of reasoning in losses for training generative models: the loss should smoothly improve as generator samples approach the target distribution, even if the supports differ. Viewing model evaluation metrics as a kind of "out-of-the-loop" loss function for training generative models – hyperparameter selection and model development focusing on variants with better evaluation metrics – suggests that these kinds of properties may be important for the problem of evaluation as well.

We thus explicitly enforce the layers of our GIN to have a controlled Lipschitz constant, similarly to e.g. spectral normalization in GAN discriminators [29]. To this end, we fix the $\lambda$ Lipschitz factor to 1.0 in the experiments. For each linear layer with weights $\mathbf{W}_\ell$, we use projected gradient descent; after each update on the weights, if $\|\mathbf{W}_\ell\| > 1.0$, we update the value of the weights to $\mathbf{W}_\ell = \frac{\mathbf{W}_\ell}{\|\mathbf{W}_\ell\|}$. This guarantees small changes in the graphs, such as adding/removing edges, or change in the input features, will not drastically change the final representation.

## 5    Experimental Results

In all of the experiments we train the model on the full dataset in an self-supervised manner. Following [46], we take the dataset and make perturbations on the dataset and see what is the trend in the

---

[2]Technically, the augmentations we use (described in Section 4) would result in a densely-connected augmentation graph: for instance, each graph has some vanishingly small probability of being reduced to an empty graph. We can instead think about the augmentation graph based on the augmentations from a high-probability set for each training example.

measurements as the perturbation degree increases. We denote perturbation degree with $r$, and define it for each type of perturbation. We use these type of perturbations:

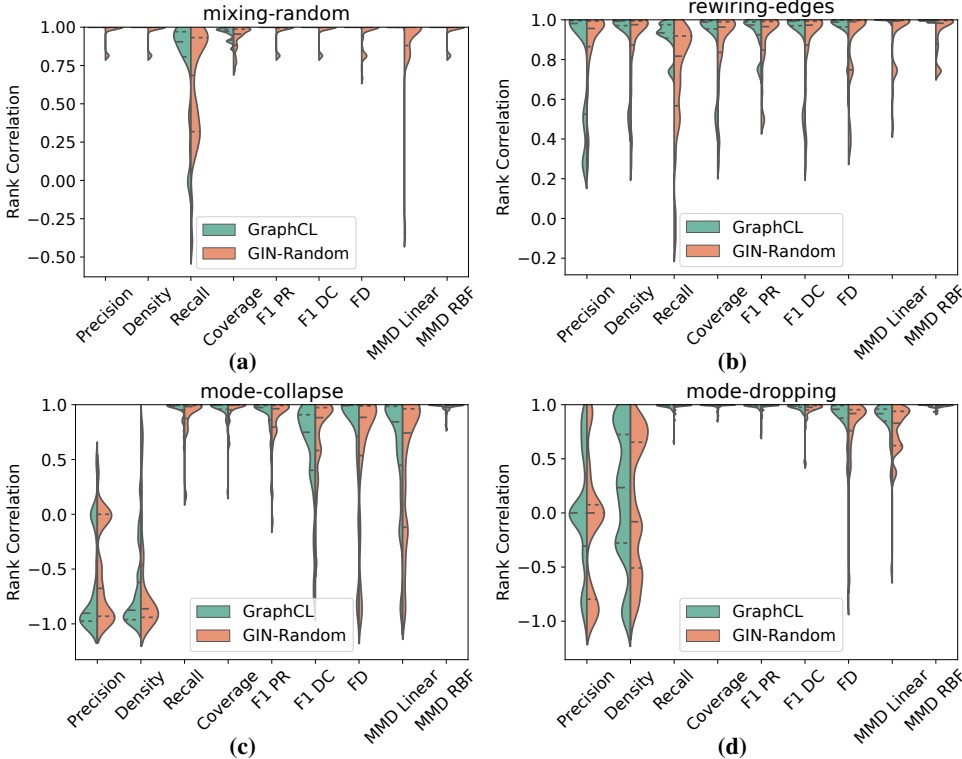

**Figure 3:** Pretrained GraphCL model versus randomly initialized GIN networks, violin results. The lines in the plots represent the quartiles. The results are gathered over all datasets and all random seeds. Self-supervised training shows better performance overall.

- **Mixing-Random:** In perturbation with ratio $r$, we remove $r$ chunk of the reference samples, and replace them with Erdős–Rényi (ER) graphs with the same ratio of edges.
- **Rewiring Edges:** This perturbation, rewires each edge of the graph with probability $r$. Each rewired edge, will change one of the sides of the edge with equal probability to another node that is not already connected to the stable node.
- **Mode Collapse and Mode Dropping:** For these perturbation, first we cluster the graphs using the WL-Kernel. First, we choose $r$ ratio of clusters, then, for mode collapse, we replace each graph on that dataset with the center of the cluster. For, mode dropping, we remove the selected clusters and then for making size of the dataset fixed, we randomly repeat samples from other clusters.

For each experiment, we measure the Spearman rand correlation between the perturbation ratio, $r$, and the value of the measurement. For measurements that supposed to decrease by the increase of ratio, we flip the values. As a result, in all experiments higher is better. We gather the results among different datasets and several random seeds and plot them for distribution of the correlations. For detailed experiments on the individual datasets see Appendix D.

**Datasets:** Following [46], we use six diverse datasets ( three synthetic and three real-world) that are frequently used in literature including: (1) **Lobster** [9] consisting of stochastic graphs with each node at most 2-hops away from a backbone path, (2) 2D **Grid** graphs [9, 25, 52], (3) **Proteins** [11] where represents connectivity among amino acids if their physical distance is less than a threshold, (4) 3-hop **Ego** networks [52] extracted from the CiteSeer network [39] representing citation-based connectivity among documents, (5) **Community** [52] graphs generated by Erdős–Rényi model [12], and (6) **Zinc** [22] is a dataset of attributed molecular graphs which allows to study sensitivity of metrics to changes

in node/edge feature distributions. We follow the exact protocols used in [9, 25, 46, 52] as follows. We randomly sample 1000 graphs from Zinc dataset and use one-hot encoding for edge/node features. For community dataset, we set $n = |\mathcal{V}|/2$, $p = 0.3$, and add $0.05|\mathcal{V}|$ inter-community edges with uniform probability. For all datasets, we use the node size range indicated in Table 4 in Appendix.

**Contrastive Training Versus Random GIN:** In the first experiment, we will examine the effects of contrastive training. We compare results of self-supervised GraphCL models versus randomly initialized GINs. For this comparison, we do not use structural features. The results are shown in Figure 3, and mean/median values are given in Table 1. In general we can see that in most measurements pretraining shows superior performance compared to the random network. Overall InfoGraph shows similar results. For InfoGraph results check Appendix D. In this experiments we use 5 random seeds and gather data on all datasets. In Appendix D, same results separated for each dataset can be seen. In our experience, pretraining shows near perfect results on larger datasets, but for Lobster and Grid datasets some correlations are not near to 1. Our observation is these measurements are moving with the correct trend up to some perturbation ratio; but for example precision/recall become zero after some perturbation and keeps still. Our intuition here is because of highly regular structures in these datasets, model learns instantly to discriminate the real graphs from the fake ones very easily. And after small amount of perturbation the perturbed graphs are all very far from the reference graphs.

**Table 1:** Mean/Median values from Figures 3 and 4. Table summarizes the distributions by their mean and median values. The results are gathered across all datasets and random seeds.

| Experiment | Model Name | Precision | Density | Recall | Coverage | F1PR | F1DC | FD | MMD Lin | MMD RBF |
|---|---|---|---|---|---|---|---|---|---|---|
| Mixing Random | GIN-Random | 0.97/**1.0** | 0.97/**1.0** | 0.59/0.69 | 0.92/0.95 | 0.97/**1.0** | 0.97/**1.0** | 0.94/**1.0** | 0.91/**1.0** | 0.97/**1.0** |
| | GraphCL | **1.0/1.0** | **1.0/1.0** | 0.77/**0.9** | 0.95/**0.97** | **1.0/1.0** | **1.0/1.0** | **1.0/1.0** | **1.0/1.0** | **1.0/1.0** |
| | GraphCL Full | **1.0/1.0** | **1.0/1.0** | **0.78**/0.89 | **0.96/0.97** | **1.0/1.0** | **1.0/1.0** | 0.97/**1.0** | 0.95/**1.0** | **1.0/1.0** |
| Rewiring Edges | GIN-Random | **0.87**/0.96 | 0.88/0.98 | 0.71/0.82 | 0.87/0.96 | 0.89/**0.97** | 0.88/0.97 | 0.87/**0.99** | 0.91/0.99 | 0.93/0.98 |
| | GraphCL | 0.79/**0.98** | **0.91/0.99** | **0.91/0.93** | **0.9/0.99** | **0.93**/0.97 | **0.91/0.99** | 0.94/**0.99** | **0.99/1.0** | **0.99/1.0** |
| | GraphCL Full | 0.86/**0.98** | 0.86/**0.99** | 0.79/0.87 | 0.87/**0.99** | 0.87/**0.99** | 0.86/**0.99** | **0.95**/0.98 | 0.97/0.99 | 0.96/0.99 |
| Mode Collapse | GIN-Random | -0.51/-0.68 | -0.61/-0.86 | 0.89/0.98 | 0.95/**1.0** | 0.86/0.96 | **0.73/0.88** | 0.57/0.88 | 0.43/0.74 | **0.98/1.0** |
| | GraphCL | -0.65/-0.9 | -0.73/-0.88 | 0.94/**0.99** | 0.94/**1.0** | 0.89/**0.97** | 0.58/0.75 | 0.71/**0.99** | 0.59/0.84 | **0.98/1.0** |
| | GraphCL Full | -0.63/-0.95 | -0.61/-0.76 | **0.95/1.0** | **0.97/1.0** | **0.91/0.97** | 0.63/0.73 | **0.75/0.99** | **0.74/0.91** | **0.98/1.0** |
| Mode Dropping | GIN-Random | -0.14/**0.0** | 0.05/-0.08 | **0.98/1.0** | **0.99/1.0** | **0.98**/0.99 | 0.94/0.98 | 0.79/0.92 | 0.77/0.83 | 0.98/0.99 |
| | GraphCL | 0.01/**0.0** | 0.15/0.23 | **0.98**/0.99 | **0.99/1.0** | **0.98/0.99** | 0.96/0.99 | 0.84/0.96 | 0.81/**0.92** | **0.99/0.99** |
| | GraphCL Full | **0.03/0.0** | **0.28/0.3** | 0.96/**1.0** | **0.99/1.0** | 0.95/**0.99** | **0.97**/0.98 | **0.89/0.97** | **0.86/0.92** | **0.99/0.99** |

**Effect of Adding Structural Information:** To conduct this experiment, we concatenated node degrees and clustering coefficients for each node feature. Pretrained GraphCL models with and without structural features are compared. Figure 4 shows the results, gathered over all datasets and random seeds; 1 shows mean/median values. Here we can see that adding these features improves the models power for some measurements. However, on highly regular and small datasets such as Grid results become poorer. Again intuition is more powerful model can easier distinguish main graphs from the fake ones. Break down of the results on the datasets in Appendix D proves this point.

**Generative Models Benchmark** In this experiment, we use samples generated from the GRAN [25] after $25\%$ of epochs, after full training, and also compare to a holdout set of the real data (half of the original dataset). We pretrain a model with full structural features and evaluate the criteria on them. As expected, the early-training results are worse than post-training results, while the real data gives almost perfect results. Results are provided in Table 2.

**Table 2:** Comparative results for training GRAN model for $25\%$ of epochs, full training, and $50/50$ split of the datasets. GGM stands for Graph Generative Model.

| Dataset | GGM | Precision↑ | Density↑ | Recall↑ | Coverage↑ | F1PR↑ | F1DC↑ | FD↓ | MMD Lin↓ | MMD RBF↓ |
|---|---|---|---|---|---|---|---|---|---|---|
| Lobster | GRAN-25% | 0.24 | 0.23 | 0.56 | 0.43 | 0.34 | 0.30 | $6.4e5$ | $3.1e4$ | 0.24 |
| | GRAN-100% | 0.38 | 0.27 | 0.75 | 0.47 | 0.50 | 0.34 | $4.7e5$ | $2.1e4$ | 0.14 |
| | 50/50 split | 1.0 | 1.0 | 0.98 | 1.0 | 0.99 | 1.0 | 3.7 | 0.53 | 0.001 |
| Grid | GRAN-25% | 0.0 | 0.0 | 1.0 | 0.0 | 0.0 | 0.0 | $3.6e9$ | $2.3e9$ | 0.94 |
| | GRAN-100% | 0.33 | 0.29 | 1.0 | 0.76 | 0.50 | 0.42 | $1.4e6$ | $1.57e5$ | 0.075 |
| | 50/50 split | 1.0 | 1.0 | 1.0 | 1.0 | 1.0 | 1.0 | 0.86 | 0.81 | 0.009 |

**Ablation Study:** We analyze how layer normalization and using subgraph augmentations in GraphCL reflects on the final results. We have used no structural feature setup and conducted

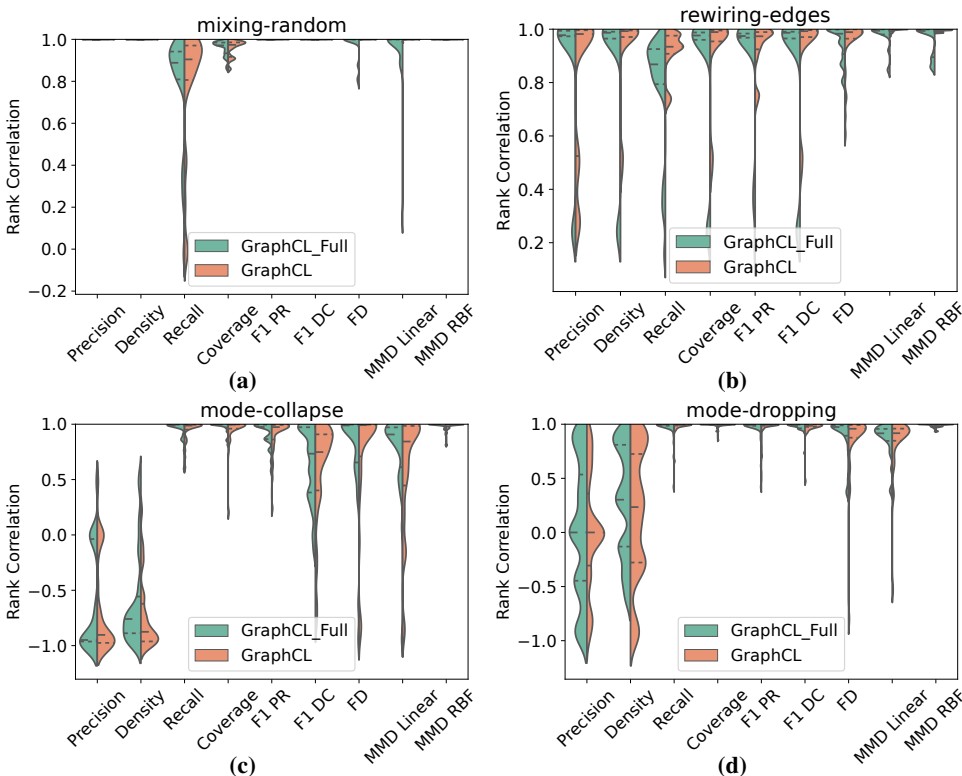

**Figure 4:** Pretrained GraphCL models with and without structural features were compared using violin results. The GraphCL model does not have structural features, while the GraphCL_Full model has structural features in addition to GraphCL. In the plots, the lines represent quartiles. The results are gathered across all datasets and random seeds. The overall distribution improves with structural features in some criteria.

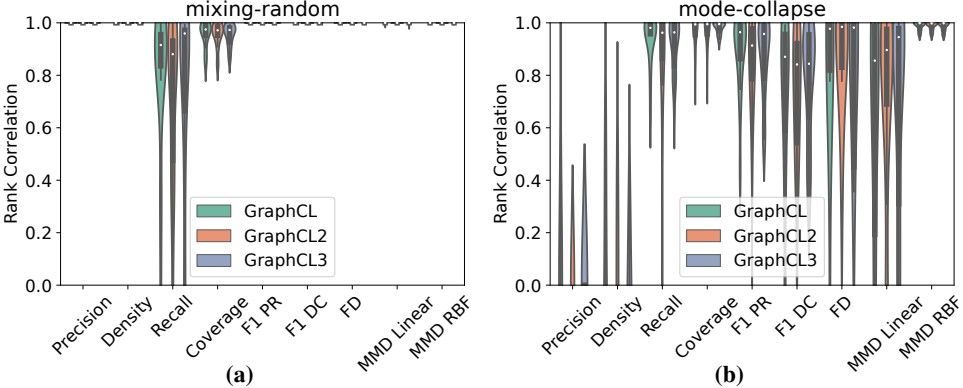

**Figure 5:** Comparison of the full method on no feature data versus removing layer normalization versus removing subgraph data augmentations on pretraining the network. GraphCL is normal training. In GraphCL2 we do not enforce lipschitzness. In GraphCL3 we remove the subgraph constraints and reduce the probability of node and edge dropping in the augmentations.

the experiment for mixing-random and mode-collapse experiments. Figure 5, shows the results on this task. The results prove that both of these improvements are essential for getting better results.

# 6 Conclusion

We have demonstrated that self-supervised pretraining of representations can yield significantly better metrics for graph evaluation than random ones, particularly when incorporating local graph features with Lipschitz control, as inspired by theory. We suggest graph generative modeling papers should consider evaluating with these metrics in addition to or instead of their existing ones.

## Acknowledgments and Disclosure of Funding

This research was enabled in part by support, computational resources, and services provided by the Canada CIFAR AI Chairs program, the Natural Sciences and Engineering Research Council of Canada, WestGrid, and the Digital Research Alliance of Canada.

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
