# A Preliminaries

We now give further description of some background material.

## A.1 Graph Neural Networks

Suppose we have a set of graphs $\mathbb{G} = \{G_1, \ldots G_N\}$ where each graph $G = (\mathcal{V}, \mathcal{E}, \mathbf{X}, \mathbf{E})$ consists of $|\mathcal{V}|$ nodes and $|\mathcal{E}|$ edges such that an edge $e_{ij} \in \mathcal{E}$ connects nodes $v_i, v_j \in \mathcal{V}$ ($\mathcal{E} \subseteq \mathcal{V} \times \mathcal{V}$). If the graph contains extra data beyond simply the connectivity structure, initial node features $\mathbf{X} \in \mathbb{R}^{|\mathcal{V}| \times d_x}$ and/or edge features $\mathbf{E} \in \mathbb{R}^{|\mathcal{V}| \times |\mathcal{V}| \times d_e}$ will also be available, where $d_x$, and $d_e$ denote the dimension of initial node and edge features, respectively. Message-passing GNNs learn low-dimensional node representations $h_v^{(k)} \in \mathbb{R}^{d_h}, \forall v \in \mathcal{V}$ as follows:

$$h_v^{(k)} = \phi \left( h_v^{(k-1)}, \bigoplus_{u \in \mathcal{N}(v)} \psi \left( h_v^{(k-1)}, h_u^{(k-1)}, e_{uv} \right) \right) \tag{2}$$

where $\mathcal{N}(v)$ is a set of immediate neighbors of node $v$, $\phi$ and $\psi$ are two arbitrary functions, and $\bigoplus$ is the aggregation function. We can use a read-out function to aggregate the learned node representations into a graph representation: $h_G = \mathcal{R}(\{h_v | v \in \mathcal{V}\})$. Different instantiations of $\phi, \psi, \bigoplus$ results in different flavors of GNNs such as message passing neural network (MPNN) [14], graph convolutional network (GCN) [23], graph attention network (GAT) [47], and graph isomorphism network (GIN) [51]. Particularly, GIN (in which $\phi$ and $\psi$ are designed as MLP and identity functions, respectively) is shown to be as powerful at detecting isomorphism as the standard Weisfeiler-Lehman (WL) test.

## A.2 Evaluating Graph Generative Models in Latent Space

Suppose we have two sets of graphs $\mathbb{G}^{train} = \{G_1, \ldots G_S\}$ and $\mathbb{G}^{test} = \{G_1, \ldots G_M\}$, each sampled from the same data distribution $p(G)$. Also suppose that we have access to an unconditional graph generative model $g_\phi(.)$, which is trained on $\mathbb{G}^{train}$ to learn the distribution of the observed set of graphs. We sample a set of generated graphs $\mathbb{G}^{gen} = \{G_1, \ldots G_N\} \sim p_{g_\phi}(G)$ from this model. In order to evaluate the quality of the sampled graphs (i.e., to decide whether the model $g_\phi(.)$ has successfully recovered the underlying distribution $p(G)$), we can define a measure of divergence $\mathcal{D}(\mathbb{G}^{test}, \mathbb{G}^{gen})$ to quantify the discrepancy between distributions of the real and generated graphs. One robust way to achieve this is to define the metric on latent vector representation spaces, and expect representations of graphs rather than the original objects. Thus, to use these metrics, we need to train a shared encoder $f_\theta(.)$ and then compute the discrepancy as $\mathcal{D}(\mathbf{H}^t, \mathbf{H}^g)$ where $\mathbf{H}^t = f_\theta(\mathbb{G}^{test}) \in \mathbb{R}^{M \times d_h}$ and $\mathbf{H}^g = f_\theta(\mathbb{G}^{gen}) \in \mathbb{R}^{N \times d_h}$. There are a few such metrics well-studied in visual domains that can differentiate the fidelity and diversity of the model, and which we can adopt in graph domains.

**Fréchet Distance (FD)** [20] models the graph representations as multivariate Gaussian distributions with mean and covariance $\mu$ and $\mathbf{C}$, and then computes the squared Wasserstein-2 distance between them: $\mathcal{D}(\mathbf{H}^t, \mathbf{H}^g) = ||\mu_t - \mu_g||_2^2 + \text{Tr}(\mathbf{C}_t + \mathbf{C}_g - 2(\mathbf{C}_t \mathbf{C}_g)^{1/2})$. FD based on Inception v3 features of images has been shown to correlate with human judgments in some settings. It can also detect intra-class mode collapse [5]. The estimator, however, has low variance but high bias [4].

**Precision & Recall (PR)** [37] measure constructs a hyper-sphere for each representation by extending a radius to its kth nearest neighbor and then aggregates the hyper-spheres to define a manifold. It then computes the precision and recall based on generated samples that fall within the manifold of real samples and vice versa. PR is also shown to correlate with human judgments in visual domain. It can also detect mode collapse and mode dropping.

**Density & Coverage (DC)** [32] is similar to PR but instead of aggregating the hyper-spheres, treats them independently. Density is computed as the average number of real hyper-spheres that a generated sample falls in and Coverage is computed as the average number of generated hyper-spheres that a real samples falls in. DC is shown to be more robust compared to PR.

**Maximum Mean Discrepancy (MMD)** [15] compares two distributions of any type, based on item-level comparison by a kernel function (e.g., polynomial kernel $\mathcal{K}(x_i, x_j) = (x_i^\top x_j + 1)^p$, Gaussian

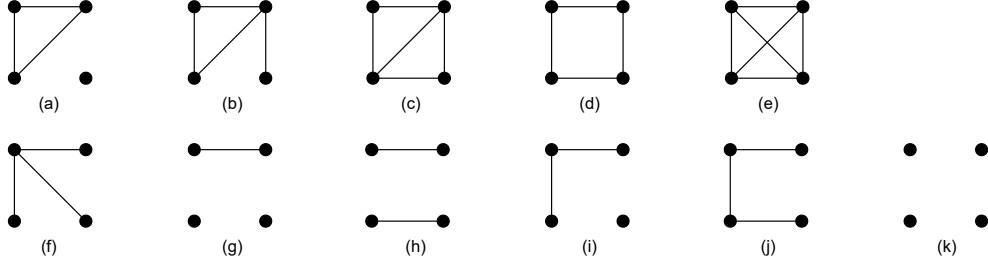

**Figure 6:** Different possible 4 node orbits.

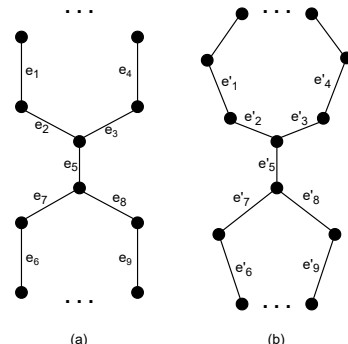

**Figure 7:** a and b are $\mathcal{C}_{a,b}$ and $\mathcal{C}_{c,d}$ respectively. We use naming from this figure for the proof.

kernel $\mathcal{K}(x_i, x_j) = \exp\left(-\frac{d(x_i,x_j)^2}{2\sigma^2}\right)$, or linear kernel $\mathcal{K}(x_i, x_j) = x_i^\top x_j$). The usual estimator is

$$\text{MMD}\left(\mathbf{H}^t, \mathbf{H}^g\right) = \frac{1}{M^2} \sum_{\substack{i,j=1 \\ i \neq j}}^{M} \mathcal{K}\left(h_i^t, h_j^t\right) + \frac{1}{N^2} \sum_{\substack{i,j=1 \\ i \neq j}}^{N} \mathcal{K}\left(h_i^g, h_j^g\right) - \frac{2}{NM} \sum_{i=1}^{N} \sum_{j=1}^{M} \mathcal{K}\left(h_i^g, h_j^t\right). \quad (3)$$

## B  Proofs

Let $\mathcal{C}_{i,j}$ denote the graph constructed by taking disjoint cycles of size $i$ and a cycle of size $j$, then connecting the two with a single edge between one node from each side.

**Proposition 1.** *For any $a, b, c, d$ satisfying $4 < a < c < d < b$ and $a + b = c + d$, $\mathcal{C}_{a,b}$ and $\mathcal{C}_{c,d}$ are not distinguishable using local metrics, but there exists a GNN that can distinguish the two.*

*Proof.* First we will prove that local metrics can not identify these two class of graphs from each other:

Both graphs have $a + b - 2$ nodes of degree 2 and 2 nodes of degree 3 so the distribution of degrees is equal in both graphs. For clustering coefficient, none of the graphs has a cycle of size 3, so it is 0 for both graphs.

We will count the number of different possible orbits of 4 nodes (all shown in Fig. 6), and show the count is the same between two graphs.

Orbits a-e occur in neither of the two graphs, so their counts are the same.

Orbit f appears just twice in both graphs.

For g and h: Consider an edge in an arbitrary graph, if we remove the edge and all nodes connected to either sides of the edge, in the remaining graph selection of each two connected nodes with the first edge will make pattern h, and two non-connected with the removed edge will make pattern g. And these are all patterns g and h that we can see in these graphs. Now, let us assume a mapping from edges of $\mathcal{C}_{a,b}$ to $\mathcal{C}_{c,d}$, by $e_i \leftrightarrow e_i'$ for the named edges in figure 7. And assume any arbitrary mapping

for other edges. If we call this mapping $f$, we can see that removing edge $e$ and all its neighbors, leaves the same number of edges and nodes as removing $f(e)$ and its neighbors for the second graph. This results as having the same number of patterns of g and h in both graphs.

For pattern i, if we assume two connected edges and again remove all the nodes connected to the nodes of these edges, any of the remaining nodes + two removed edges will shape pattern i. Again by the same mapping $f$, we can see that removing any of these two connected edges from the first graph leaves the same number of nodes as removing from the second graph.

For pattern j, we can divide these patterns into two groups: 1) Patterns that do not include $e_5$ or $e_5'$. 2) Patterns that include these edges. For the first 1, we can assume we have two separate cycles on each of the graphs. Each cycle of size $n$ has $n$ of these patterns, so first graph has a total of $a + b$ of these patterns, and the second graph has a total of $c + d = a + b$ of these patterns. So, these class of patterns have same amount of repetition in two graphs. For (2), we can easily see from the 7 there are 8 of them in each of the graphs. So, the number of patterns of this type is also equal in both graphs.

For pattern k, as two graphs have same number of nodes, we have $\binom{n}{k}$ possible selection of four nodes, and as the number of patterns of type a-j is equal between the graphs, pattern k should also have same number of repeat in both graphs.

As a result, the distribution of local metrics is exactly same between two classes of graphs, and thus these two graphs can not be distinguished using these metrics.

Now, we prove that with a deep enough GNN, and good set of weights, we can distinguish these two graphs. Assume all nodes have no information more than their degrees, at first all of the nodes except two sides of bridge in each graph have the same message. There are two messages of degree 3, and rest of degree 2. In cycle with size $a$, after $a$ steps, the initial node with degree 3, will finally receive a message that has started from itself, and so it would be different from any message that are already spread in 3 other circles. As a result, there are at least two nodes with different messages between two graphs after $a$ steps. Thus two graphs are distinguishable using this GNN. □

For Fig. 1 (b), it is well-known that these two graphs can not be identified using WL-test, and thus results of Xu et al. [51] imply that GNNs of order 1 can not distinguish these two graphs. On the other hand, for local structures, one of the graphs has two triangles and the other one has none. As a result clustering coefficients of the nodes will be different.

## C Implementation Details and Hyperparameters

In addition to the details here, code is available in the supplementary materials, and will be made public with release of the paper.

### C.1 Training Self-Supervised Methods

For training GraphCL networks, we use node dropping and edge dropping probabilities of $0.1$ for both. We use random walks of length 10 as subgraphs. We do not use node feature masking augmentations; here essentially all the datasets except for the ZINC do not have any node features, and we use constant value of $1.0$ for their node features. Removing just constant would leave some nodes without any features. For the experiments with structural features, the point was to see how the structural features work, as a result again we do not use feature dropping here as well. During the training, we use structural features as default node features; this means that after augmentations like node/edge dropping, we do not recalculate the features for the new graph. This makes training process faster, but can affect the learning process in GraphCL only.

The hyperparameters used for GINs are provided in 3. These hyperparameters are shared among GraphCL, InfoGraph and Random GIN models. For other hyperparameters of the self-supervised training we follow the instructions from Sun et al. [42] and You et al. [54]. We train the self-supervised methods and evaluate them based on their criteria loss. After finding proper hyperparameters for that dataset, we train the model with those hyperparameters and among all of the experiments for that dataset we use the same model (no retraining for each experiment).

**Table 3:** Hyperparameters of the networks.

|  | **Lobster** | **Grid** | **Community** | **Ego** | **Proteins** | **ZINC** |
|---|---|---|---|---|---|---|
| Number of GIN Layers | 2 | 3 | 3 | 3 | 3 | 3 |
| GIN Hidden Layer Size | 16 | 16 | 32 | 32 | 32 | 32 |
| Epochs | 40 | 20 | 100 | 100 | 100 | 100 |
| Lipchitz Factor | 1.0 | 1.0 | 1.0 | 1.0 | 1.0 | 1.0 |

## C.2 Random GINs

Experiments in [46] show that results on random GINs tend to have very small variance among different setups of the structures of the GINs. For fair evaluation, we used the same GIN hyperparameters among all of the models for the same dataset. We use batch norm and orthogonal initialization as instructed in the main paper. For each experiment with a different seed, we use a different randomly initialized GIN. For experiments with structural features, we give the same features as input for the Random GIN as well. Random GIN with no structural features will be the same as the method proposed in [46].

## C.3 Benchmark Experiment Setups

We follow the exact instruction and public code released for [46] for the benchmarks. Step sizes for increasing the $r$ is set to $0.01$. For mixing random we remove $r$ ratio of the graphs and for each Graph $G = (V, E)$ that has been removed we replace it with and Erdős–Rényi (ER) graph with same number of nodes and probability of connection as $p = \frac{|E|}{\binom{|V|}{2}}$. Rewiring probabilities are increasing from $0.0$ to $1.0$ with step size of $0.01$. The number of steps for mode-dropping/mode-collapse experiments is as the number of clusters found by the clustering algorithm. Experiments for no structural features and with degree features are conducted with ten random seed for each dataset. For the clustering features we have used three random seed per dataset. Ablation studies are done using $5$ random seeds per dataset.

# D   Further Experiment Results

## D.1 Comparing Contrastive Methods, GraphCL Versus InfoGraph:

As mentioned two methods for contrastive learning on graphs have been used in this paper. No specific augmentation type is used in InfoGraph, instead, local node embeddings are contrasted with graph-level embeddings. While this simplifies the choice of hyperparameters, GraphCL has more control over how augmentations are applied for domain-specific tasks. An experimental study of the InfoGraph model is presented here. All results indicate that no one method is superior to another, but since GraphCL allows for domain-specific augmentations and can be used for a variety of tasks, we have chosen it as the main method. We do not use any structural features in this set of experiments.

## D.2 Break Down of Experiments on Single Datasets

Table 4 gives details of the datasets. Figs. 9, 11, 13, 15, 17 and 19 break down the experiments with no structural features on individual datasets and Figs. 10, 12, 14, 16, 18 and 20 those with structural features. Tables 5 to 10 compare numerical values of means and medians from these plots.

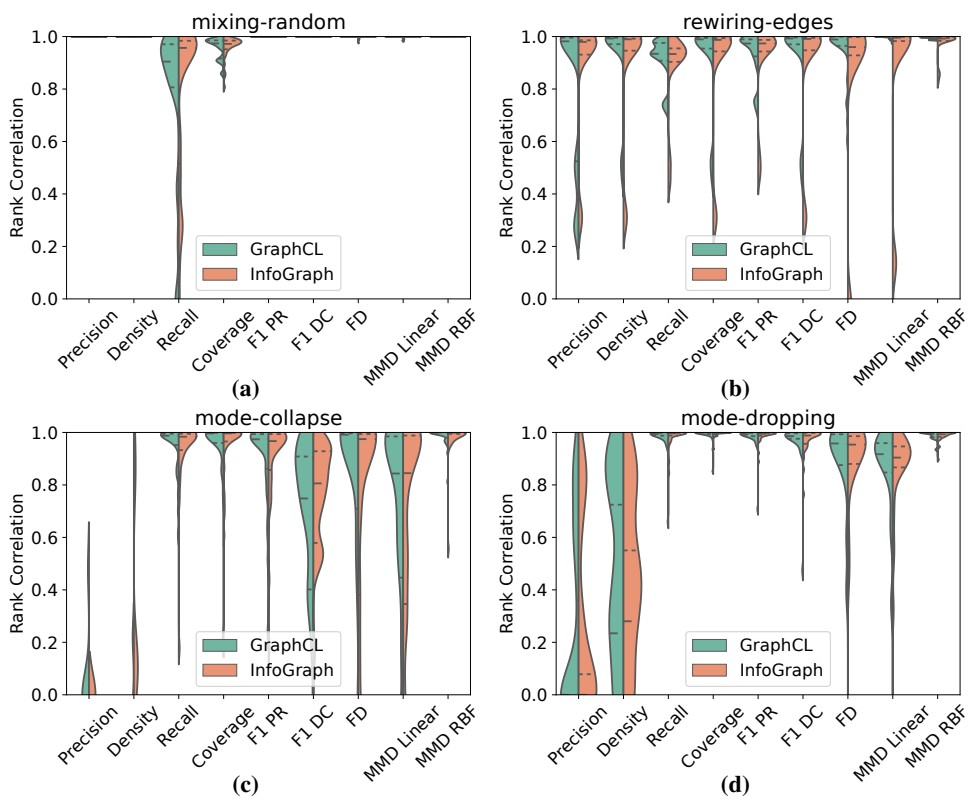

**Figure 8:** Experimental results for the InfoGraph versus GraphCL without structural features, aggregated among all datasets.

**Table 4:** Statistics of graph datasets.

| | *Synthetic* | | | *Real-World* | | |
| | **LOBSTER** | **GRID** | **COMMUNITY** | **EGO** | **PROTEINS** | **ZINC** |
|---|---|---|---|---|---|---|
| \|GRAPHS\| | 100 | 100 | 500 | 757 | 918 | 1000 |
| \|NODES\| | 10-100 | 100-400 | 60-160 | 50-399 | 100-500 | 10-50 |
| \|EDGES\| | 10-100 | 360-1368 | 300-1800 | 57-1071 | 186-1575 | 22-82 |
| \|NODE FEATURES\| | 0 | 0 | 0 | 0 | 0 | 28 |
| \|EDGE FEATURES\| | 0 | 0 | 0 | 0 | 0 | 4 |

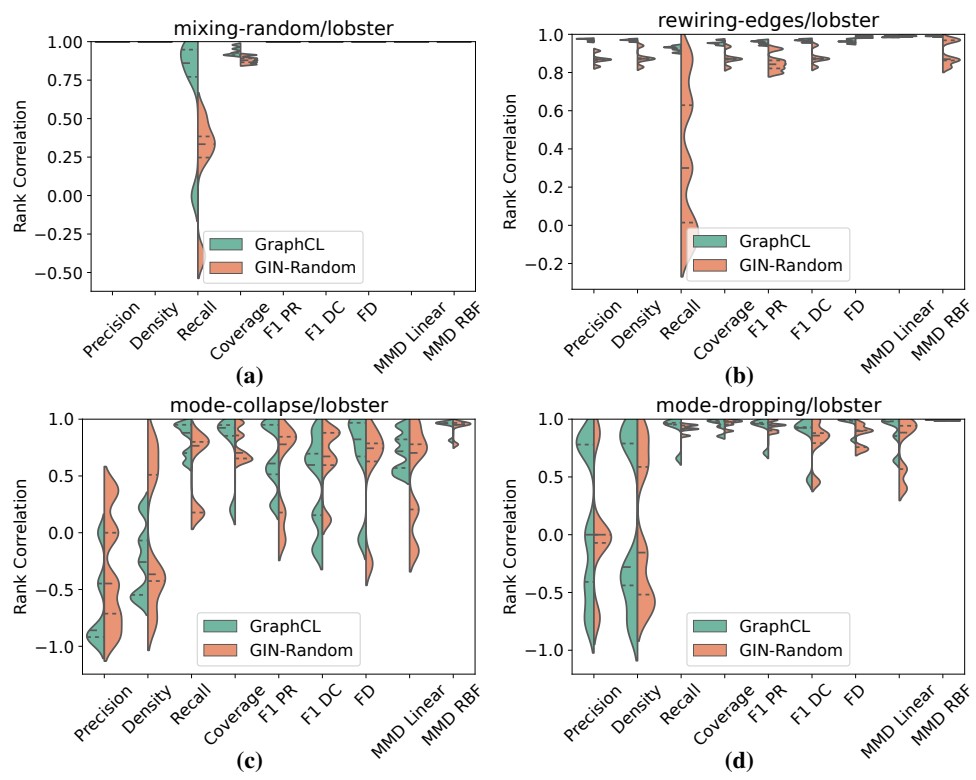

**Figure 9:** Violin comparative results among the methods, with no structural features for Lobster dataset.

**Table 5:** Comparison of the main models' mean and median experiment results on the Lobster dataset.

| Experiment | Model Name | Precision | Density | Recall | Coverage | F1PR | F1DC | FD | MMD Lin | MMD RBF |
|---|---|---|---|---|---|---|---|---|---|---|
| Mixing Random | GIN-Random | **1.0/1.0** | **1.0/1.0** | 0.22/0.33 | 0.88/0.88 | **1.0/1.0** | **1.0/1.0** | **1.0/1.0** | **1.0/1.0** | **1.0/1.0** |
| | GraphCL | **1.0/1.0** | **1.0/1.0** | 0.71/0.86 | **0.94/0.93** | **1.0/1.0** | **1.0/1.0** | **1.0/1.0** | **1.0/1.0** | **1.0/1.0** |
| | GraphCL Full | **1.0/1.0** | **1.0/1.0** | **0.9/0.93** | 0.92/0.91 | **1.0/1.0** | **1.0/1.0** | **1.0/1.0** | **1.0/1.0** | **1.0/1.0** |
| Rewiring Edges | GIN-Random | 0.87/0.87 | 0.88/0.87 | 0.34/0.3 | 0.87/0.87 | 0.85/0.84 | 0.88/0.87 | **0.99/0.99** | **0.99/0.99** | 0.9/0.87 |
| | GraphCL | **0.97/0.98** | **0.97/0.97** | **0.93/0.93** | 0.95/0.95 | 0.96/0.96 | **0.97/0.97** | 0.96/0.96 | **0.99/0.99** | **0.99/0.99** |
| | GraphCL Full | **0.97**/0.97 | 0.96/0.96 | 0.9/0.89 | **0.96/0.96** | **0.97/0.97** | **0.97**/0.96 | 0.95/0.95 | **0.99/0.99** | **0.99/0.99** |
| Mode Collapse | GIN-Random | -0.34/-0.45 | -0.05/-0.37 | 0.55/0.77 | 0.76/0.7 | 0.53/0.78 | **0.63/0.67** | 0.56/0.74 | 0.5/0.7 | 0.92/0.95 |
| | GraphCL | -0.63/-0.86 | -0.25/-0.26 | **0.82/0.88** | 0.78/0.92 | 0.65/0.61 | 0.44/0.6 | 0.68/**0.82** | 0.71/0.72 | **0.94/0.96** |
| | GraphCL Full | -0.32/-0.15 | -0.46/-0.52 | 0.81/0.86 | **0.88**/0.85 | **0.76/0.86** | 0.2/0.04 | **0.72**/0.65 | **0.79/0.84** | 0.93/**0.96** |
| Mode Dropping | GIN-Random | 0.02/**0.0** | 0.03/-0.15 | **0.92**/0.92 | 0.96/**0.98** | **0.93**/0.95 | 0.8/0.86 | 0.85/0.9 | 0.76/0.88 | **0.99**/0.99 |
| | GraphCL | 0.09/**0.0** | 0.04/-0.28 | 0.91/**0.96** | 0.95/0.97 | 0.92/**0.96** | 0.85/0.93 | **0.96/0.99** | 0.89/0.98 | **0.99/1.0** |
| | GraphCL Full | **0.16/0.0** | **0.21**/-0.12 | 0.76/0.83 | **0.96**/0.96 | 0.76/0.86 | **0.93/0.98** | 0.95/0.97 | 0.86/0.91 | **0.99/1.0** |

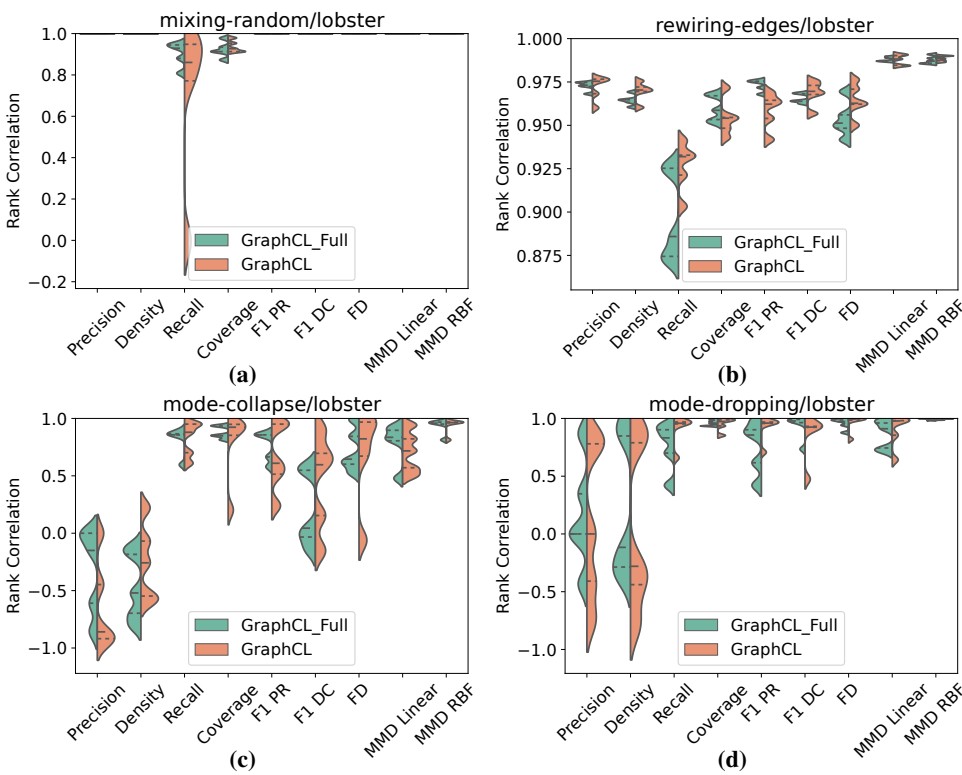

**Figure 10:** Violin comparative results among the methods, with clustering structural features for the lobster dataset.

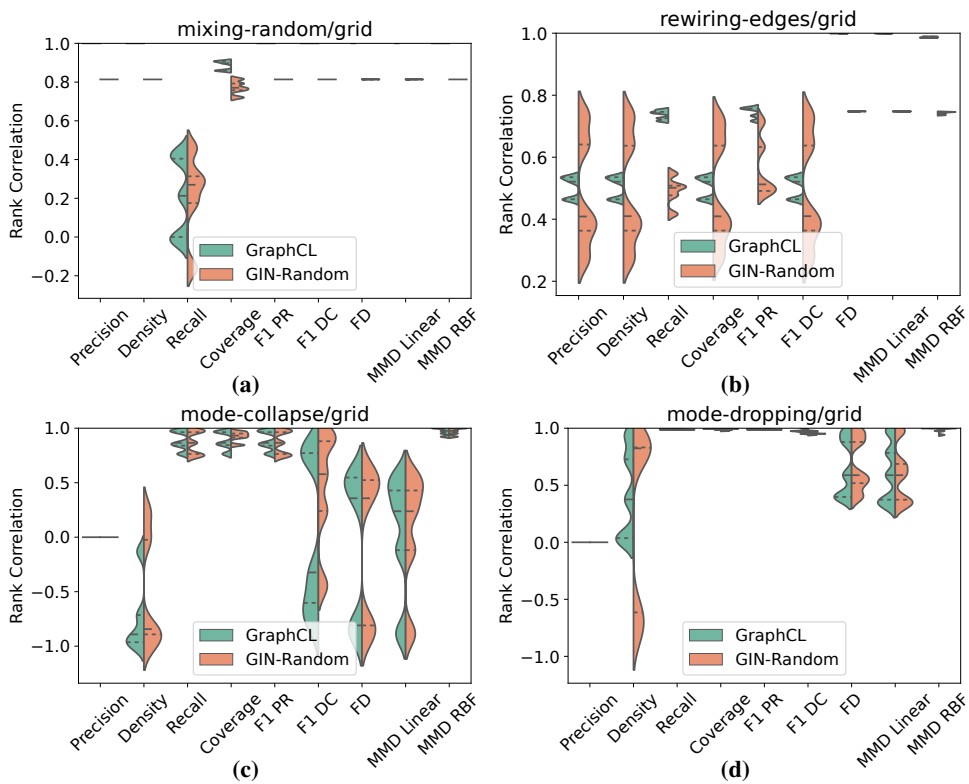

**Figure 11:** Violin comparative results among the methods, with no structural features for Grid dataset.

**Table 6:** Comparison of the main models' mean and median experiment results on the Grid dataset.

| Experiment | Model Name | Precision | Density | Recall | Coverage | F1PR | F1DC | FD | MMD Lin | MMD RBF |
|---|---|---|---|---|---|---|---|---|---|---|
| Mixing Random | GIN-Random | 0.81/0.81 | 0.81/0.81 | 0.21/0.27 | 0.77/0.77 | 0.81/0.81 | 0.81/0.81 | 0.81/0.81 | 0.81/0.81 | 0.81/0.81 |
| | GraphCL | **1.0/1.0** | **1.0/1.0** | 0.21/0.21 | 0.89/0.9 | **1.0/1.0** | **1.0/1.0** | **1.0/1.0** | **1.0/1.0** | **1.0/1.0** |
| | GraphCL Full | **1.0/1.0** | **1.0/1.0** | **0.25/0.29** | **0.91/0.92** | **1.0/1.0** | **1.0/1.0** | **1.0/1.0** | **1.0/1.0** | **1.0/1.0** |
| Rewiring Edges | GIN-Random | 0.48/0.41 | 0.48/0.41 | 0.49/0.5 | 0.48/0.41 | 0.57/0.51 | 0.48/0.41 | 0.75/0.75 | 0.75/0.75 | 0.74/0.74 |
| | GraphCL | **0.5/0.52** | **0.5/0.52** | **0.74/0.74** | **0.5/0.52** | **0.74/0.76** | **0.5/0.52** | **1.0/1.0** | **1.0/1.0** | **0.99/0.99** |
| | GraphCL Full | 0.24/0.24 | 0.24/0.24 | 0.32/0.33 | 0.24/0.24 | 0.34/0.34 | 0.24/0.24 | **1.0/1.0** | **1.0/1.0** | 0.89/0.89 |
| Mode Collapse | GIN-Random | **0.0/0.0** | -0.5/-0.84 | 0.86/0.87 | 0.92/0.93 | 0.86/0.87 | 0.44/0.58 | -0.05/**0.36** | 0.04/0.24 | **0.97/0.98** |
| | GraphCL | **0.0/0.0** | -0.74/-0.89 | 0.88/0.87 | 0.88/0.87 | 0.88/0.87 | -0.03/-0.32 | -0.05/**0.36** | 0.04/0.24 | **0.97/0.98** |
| | GraphCL Full | **0.0/0.0** | 0.06/0.17 | **0.9/0.91** | **0.95/0.97** | **0.9/0.91** | **0.75/0.77** | -0.06/0.33 | **0.17/0.4** | 0.96/0.95 |
| Mode Dropping | GIN-Random | **0.0/0.0** | 0.22/**0.82** | **0.99/0.99** | **0.99/0.99** | **0.99/0.99** | 0.96/0.95 | 0.67/0.59 | 0.59/**0.59** | 0.98/**1.0** |
| | GraphCL | **0.0/0.0** | 0.42/0.38 | **0.99/0.99** | **0.99/0.99** | **0.99/0.99** | **0.97/0.98** | 0.65/0.59 | 0.61/**0.59** | **1.0/1.0** |
| | GraphCL Full | **0.0/0.0** | **0.58**/0.77 | 0.98/**0.99** | **0.99/0.99** | 0.98/**0.99** | 0.96/0.95 | **0.7/0.76** | 0.62/0.59 | 0.99/**1.0** |

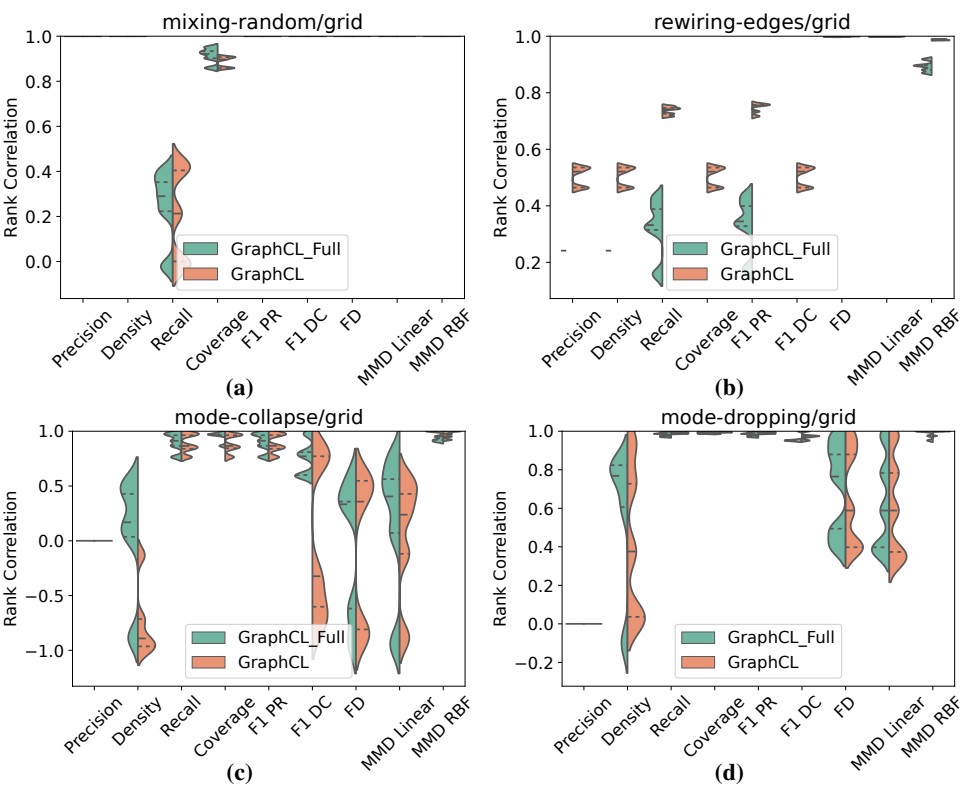

**Figure 12:** Violin comparative results among the methods, with clustering structural features for Grid dataset.

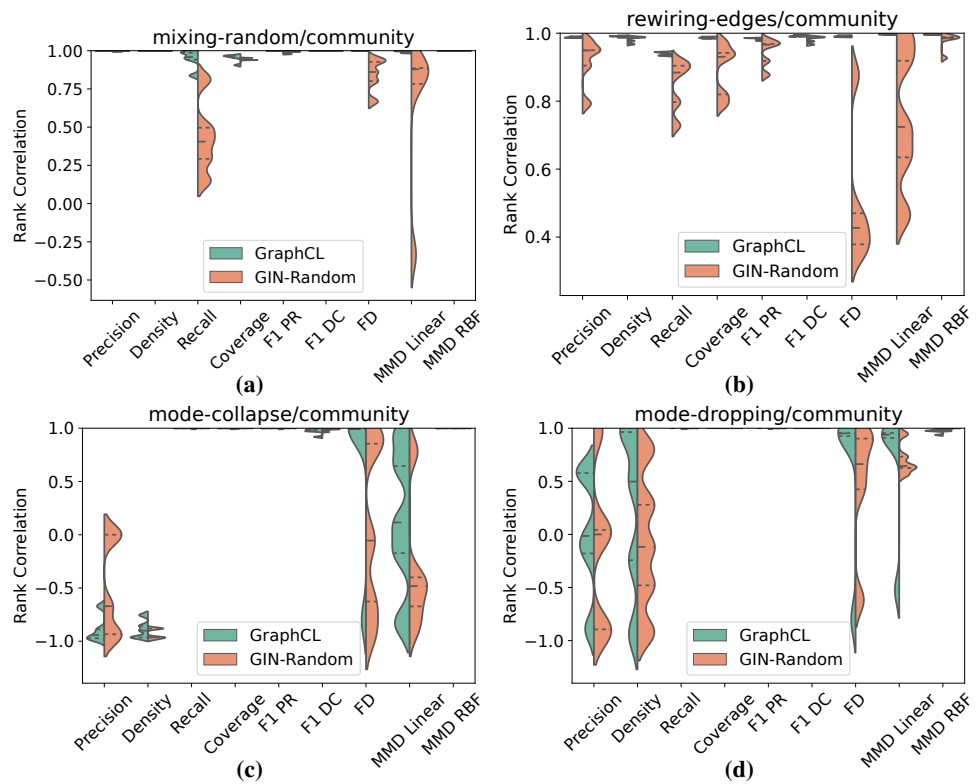

**Figure 13:** Violin comparative results among the methods, with no structural features for community dataset.

**Table 7:** Comparison of the main models' mean and median experiment results on the Community dataset.

| Experiment | Model Name | Precision | Density | Recall | Coverage | F1PR | F1DC | FD | MMD Lin | MMD RBF |
|---|---|---|---|---|---|---|---|---|---|---|
| Mixing Random | GIN-Random | **1.0/1.0** | **1.0/1.0** | 0.43/0.41 | 0.94/0.94 | 0.99/0.99 | **1.0/1.0** | 0.84/0.86 | 0.62/0.88 | **1.0/1.0** |
| | GraphCL | **1.0/1.0** | **1.0/1.0** | 0.94/0.96 | 0.95/0.96 | **1.0/1.0** | **1.0/1.0** | **1.0/1.0** | **0.99/0.99** | **1.0/1.0** |
| | GraphCL Full | **1.0/1.0** | **1.0/1.0** | **0.97/0.98** | **0.97/0.97** | **1.0/1.0** | **1.0/1.0** | 0.83/0.81 | 0.74/0.88 | **1.0/1.0** |
| Rewiring Edges | GIN-Random | 0.92/0.95 | 0.98/**0.99** | 0.85/0.88 | 0.89/0.93 | 0.94/0.97 | 0.98/**0.99** | 0.5/0.43 | 0.75/0.72 | 0.98/0.99 |
| | GraphCL | **0.99/0.99** | **0.99/0.99** | **0.94/0.94** | **0.99/0.99** | **0.98/0.99** | **0.99/0.99** | **0.99/0.99** | **1.0/1.0** | **1.0/1.0** |
| | GraphCL Full | 0.98/0.98 | **0.99/0.99** | 0.92/0.92 | 0.98/0.98 | **0.98**/0.98 | **0.99/0.99** | 0.85/0.82 | 0.88/0.85 | 0.86/0.86 |
| Mode Collapse | GIN-Random | -0.51/-0.67 | -0.93/-0.96 | **1.0/1.0** | **1.0/1.0** | **1.0/1.0** | 0.98/**0.99** | 0.03/-0.05 | -0.33/-0.48 | **1.0/1.0** |
| | GraphCL | -0.89/-0.94 | -0.89/-0.9 | **1.0/1.0** | **1.0/1.0** | **1.0/1.0** | 0.97/0.97 | 0.64/**0.99** | 0.14/0.12 | **1.0/1.0** |
| | GraphCL Full | -0.96/-0.96 | -0.76/-0.79 | **1.0/1.0** | **1.0/1.0** | **1.0/1.0** | **0.98/0.99** | **0.99/0.99** | **0.77/0.92** | **1.0/1.0** |
| Mode Dropping | GIN-Random | -0.16/0.0 | -0.09/-0.12 | **1.0/1.0** | **1.0/1.0** | **1.0/1.0** | **1.0/1.0** | 0.46/0.66 | 0.7/0.64 | **0.99/0.99** |
| | GraphCL | 0.02/-0.01 | **0.25/0.5** | **1.0/1.0** | **1.0/1.0** | **1.0/1.0** | **1.0/1.0** | 0.6/0.95 | 0.65/0.94 | 0.97/0.98 |
| | GraphCL Full | **0.77/0.9** | 0.21/0.4 | **1.0/1.0** | **1.0/1.0** | **1.0/1.0** | **1.0/1.0** | **0.99/0.98** | **0.94/0.95** | 0.98/**0.99** |

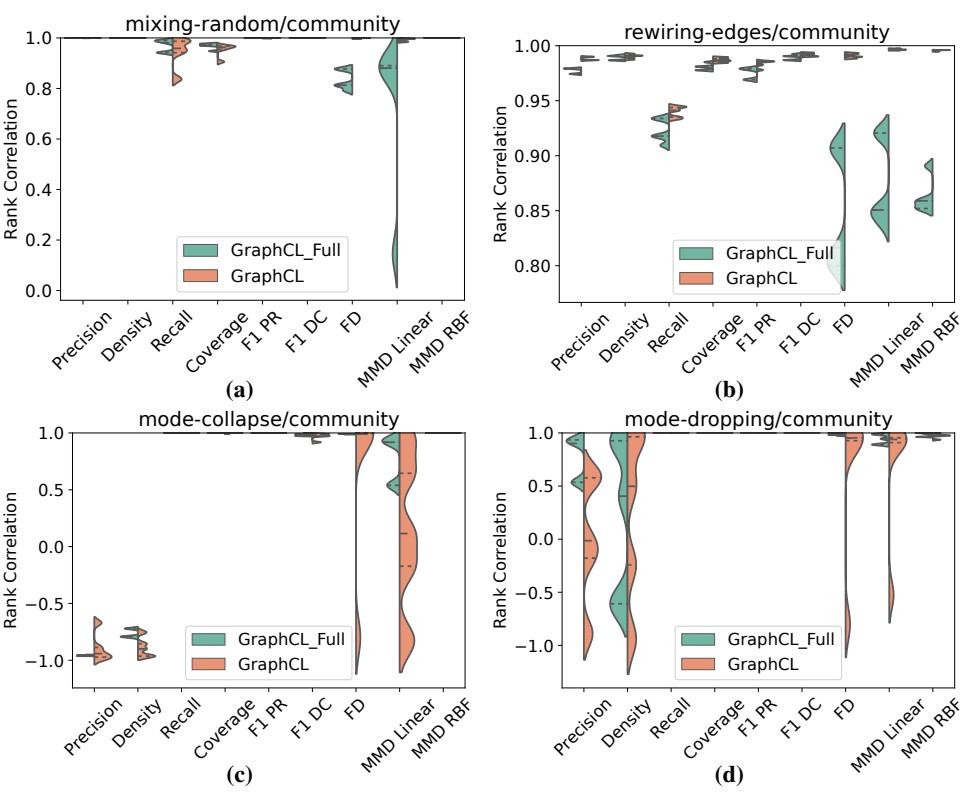

**Figure 14:** Violin comparative results among the methods, with clustering structural features for community dataset.

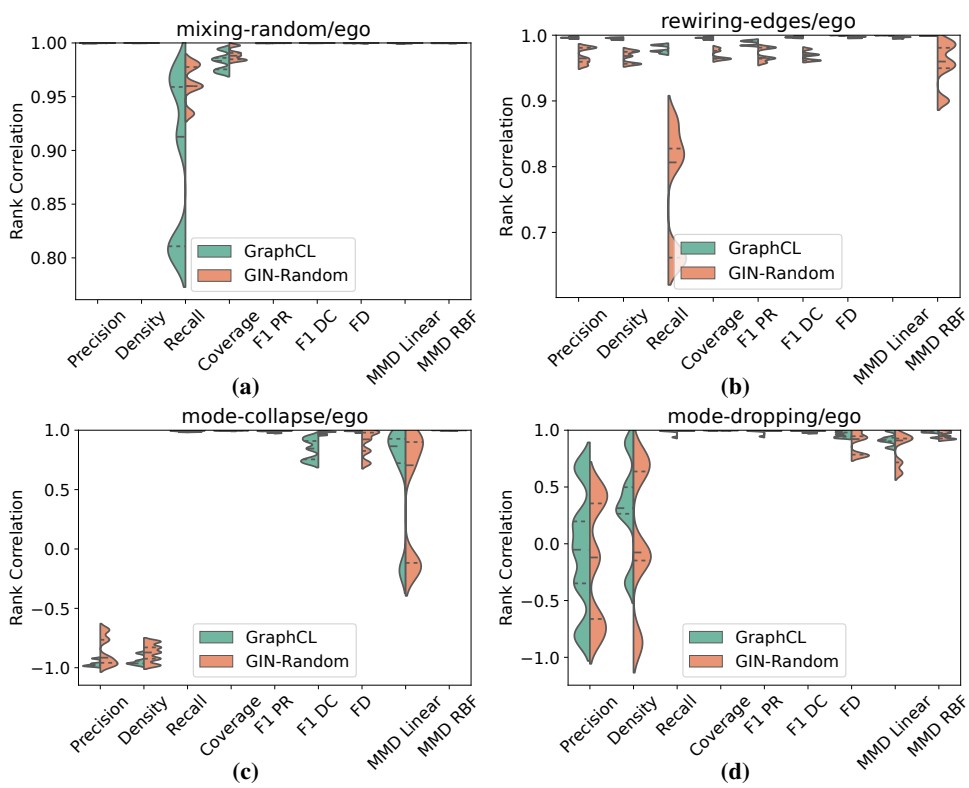

**Figure 15:** Violin comparative results among the methods, with no structural features for ego dataset.

**Table 8:** Comparison of the main models' mean and median experiment results on the Ego dataset.

| Experiment | Model Name | Precision | Density | Recall | Coverage | F1PR | F1DC | FD | MMD Lin | MMD RBF |
|---|---|---|---|---|---|---|---|---|---|---|
| Mixing Random | GIN-Random | **1.0/1.0** | **1.0/1.0** | **0.96/0.96** | **0.99/0.99** | **1.0/1.0** | **1.0/1.0** | **1.0/1.0** | **1.0/1.0** | **1.0/1.0** |
| | GraphCL | **1.0/1.0** | **1.0/1.0** | 0.89/0.91 | 0.98/0.98 | **1.0/1.0** | **1.0/1.0** | **1.0/1.0** | **1.0/1.0** | **1.0/1.0** |
| | GraphCL Full | **1.0/1.0** | **1.0/1.0** | 0.87/0.88 | 0.98/0.98 | **1.0/1.0** | **1.0/1.0** | **1.0/1.0** | **1.0/1.0** | **1.0/1.0** |
| Rewiring Edges | GIN-Random | 0.97/0.96 | 0.97/0.97 | 0.76/0.81 | 0.97/0.97 | 0.97/0.97 | 0.97/0.97 | **1.0/1.0** | **1.0/1.0** | 0.96/0.96 |
| | GraphCL | **1.0/1.0** | **1.0/1.0** | **0.98/0.98** | **1.0/1.0** | **0.99/0.99** | **1.0/1.0** | **1.0/1.0** | **1.0/1.0** | **1.0/1.0** |
| | GraphCL Full | **1.0/1.0** | **1.0/1.0** | 0.95/0.95 | 0.99/0.99 | 0.98/0.98 | **1.0/1.0** | 0.89/0.87 | 0.98/0.98 | **1.0/1.0** |
| Mode Collapse | GIN-Random | -0.86/-0.92 | -0.88/-0.87 | 0.99/0.99 | **1.0/1.0** | **0.99/1.0** | **0.98/0.98** | 0.89/0.92 | 0.45/0.7 | **1.0/1.0** |
| | GraphCL | -0.97/-0.98 | -0.94/-0.96 | 0.99/0.99 | **1.0/1.0** | **0.99/0.99** | 0.84/0.85 | **0.99/0.99** | 0.66/0.86 | **1.0/1.0** |
| | GraphCL Full | -0.56/-0.74 | -0.94/-0.95 | **1.0/1.0** | **1.0/1.0** | **0.99/1.0** | 0.93/**0.98** | **0.99/1.0** | 0.8/0.92 | **1.0/1.0** |
| Mode Dropping | GIN-Random | -0.15/-0.12 | 0.06/-0.08 | **1.0/1.0** | **1.0/1.0** | **1.0/1.0** | **0.99/0.99** | 0.88/0.92 | 0.83/0.91 | 0.95/0.95 |
| | GraphCL | -0.07/-0.05 | 0.32/0.31 | 0.98/0.99 | **1.0/1.0** | 0.99/0.99 | **0.99/0.99** | 0.96/0.96 | 0.92/0.91 | **0.98/0.98** |
| | GraphCL Full | -0.3/-0.5 | **0.62/0.64** | 0.99/**1.0** | **1.0/1.0** | 0.99/0.99 | **0.99/1.0** | 0.97/**0.98** | **0.94**/0.92 | 0.98/**0.99** |

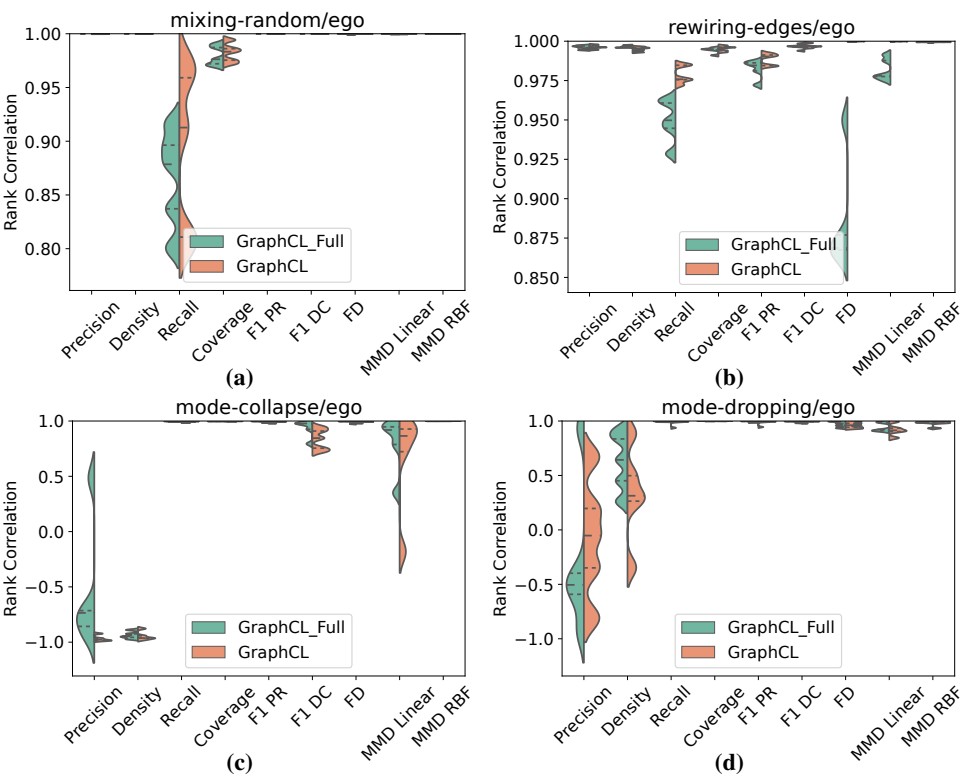

**Figure 16:** Violin comparative results among the methods, with clustering structural features for ego dataset.

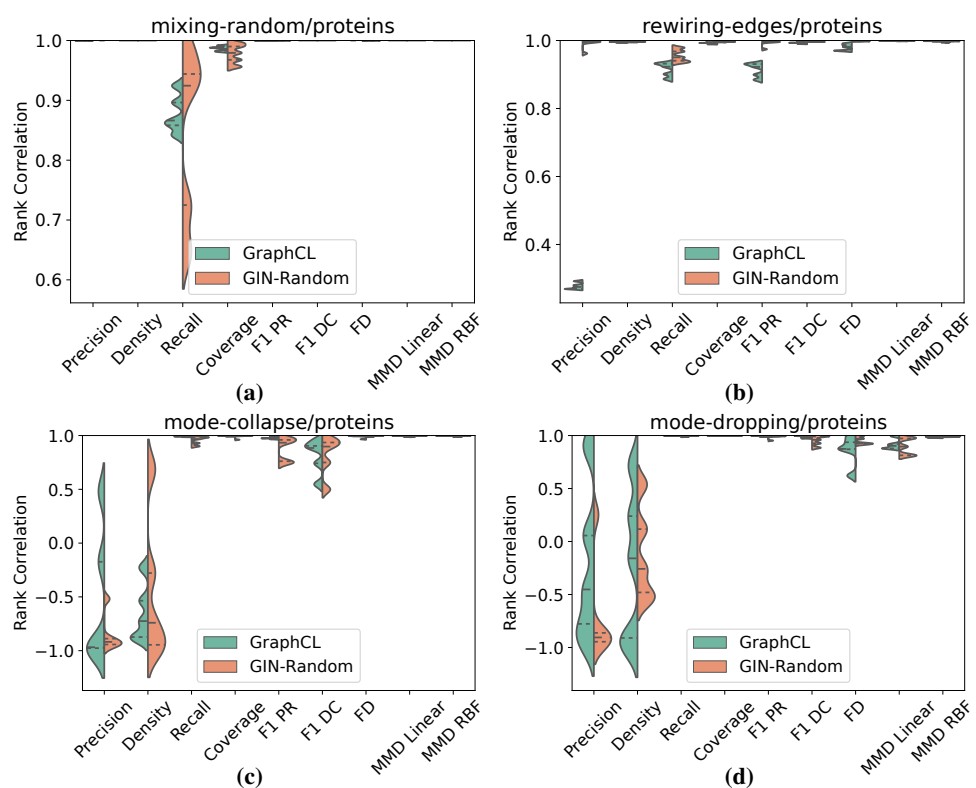

**Figure 17:** Violin comparative results among the methods, with no structural features for proteins dataset.

**Table 9:** Comparison of the main models' mean and median experiment results on the Proteins dataset.

| Experiment | Model Name | Precision | Density | Recall | Coverage | F1PR | F1DC | FD | MMD Lin | MMD RBF |
|---|---|---|---|---|---|---|---|---|---|---|
| Mixing Random | GIN-Random | **1.0/1.0** | **1.0/1.0** | 0.85/**0.92** | 0.98/0.98 | **1.0/1.0** | **1.0/1.0** | **1.0/1.0** | **1.0/1.0** | **1.0/1.0** |
| | GraphCL | **1.0/1.0** | **1.0/1.0** | **0.88**/0.87 | **0.99/0.99** | **1.0/1.0** | **1.0/1.0** | **1.0/1.0** | **1.0/1.0** | **1.0/1.0** |
| | GraphCL Full | **1.0/1.0** | **1.0/1.0** | 0.87/0.84 | **0.99/0.99** | **1.0/1.0** | **1.0/1.0** | **1.0/1.0** | **1.0/1.0** | **1.0/1.0** |
| Rewiring Edges | GIN-Random | **0.99/1.0** | **1.0/1.0** | **0.95/0.95** | **1.0/1.0** | **0.99/1.0** | **1.0/1.0** | **1.0/1.0** | **1.0/1.0** | **1.0/1.0** |
| | GraphCL | 0.28/0.28 | 0.99/**1.0** | 0.92/0.92 | 0.99/0.99 | 0.91/0.92 | 0.99/0.99 | 0.98/0.98 | **1.0/1.0** | **1.0/1.0** |
| | GraphCL Full | 0.98/0.98 | 0.98/0.98 | 0.82/0.8 | 0.97/0.97 | 0.96/0.97 | 0.98/0.98 | **1.0/1.0** | **1.0/1.0** | **1.0/1.0** |
| Mode Collapse | GIN-Random | -0.84/-0.92 | -0.45/-0.74 | 0.95/0.97 | 0.99/**1.0** | 0.88/0.93 | **0.81/0.9** | **0.99/1.0** | **0.99/1.0** | 0.99/**1.0** |
| | GraphCL | -0.53/-0.97 | -0.65/-0.72 | **0.99/0.99** | **1.0/1.0** | **0.98**/0.97 | **0.81**/0.88 | **0.99**/0.99 | **1.0/1.0** | **1.0/1.0** |
| | GraphCL Full | -0.97/-0.96 | -0.72/-0.73 | **0.99/0.99** | **1.0/1.0** | 0.96/**0.98** | 0.65/0.73 | 0.87/**1.0** | 0.95/0.99 | **1.0/1.0** |
| Mode Dropping | GIN-Random | -0.68/-0.91 | -0.13/-0.26 | 0.99/**1.0** | **1.0/1.0** | 0.99/0.99 | 0.95/0.96 | **0.95/0.93** | 0.89/**0.9** | **0.99/0.99** |
| | GraphCL | -0.25/-0.45 | -0.22/-0.16 | **1.0/1.0** | **1.0/1.0** | **1.0/1.0** | **0.99**/0.98 | 0.86/0.87 | **0.91/0.9** | **0.99/0.99** |
| | GraphCL Full | -0.17/-0.45 | -0.02/**0.26** | **1.0/1.0** | **1.0/1.0** | **1.0/1.0** | **0.99/0.99** | 0.71/0.87 | 0.87/0.87 | 0.98/0.97 |

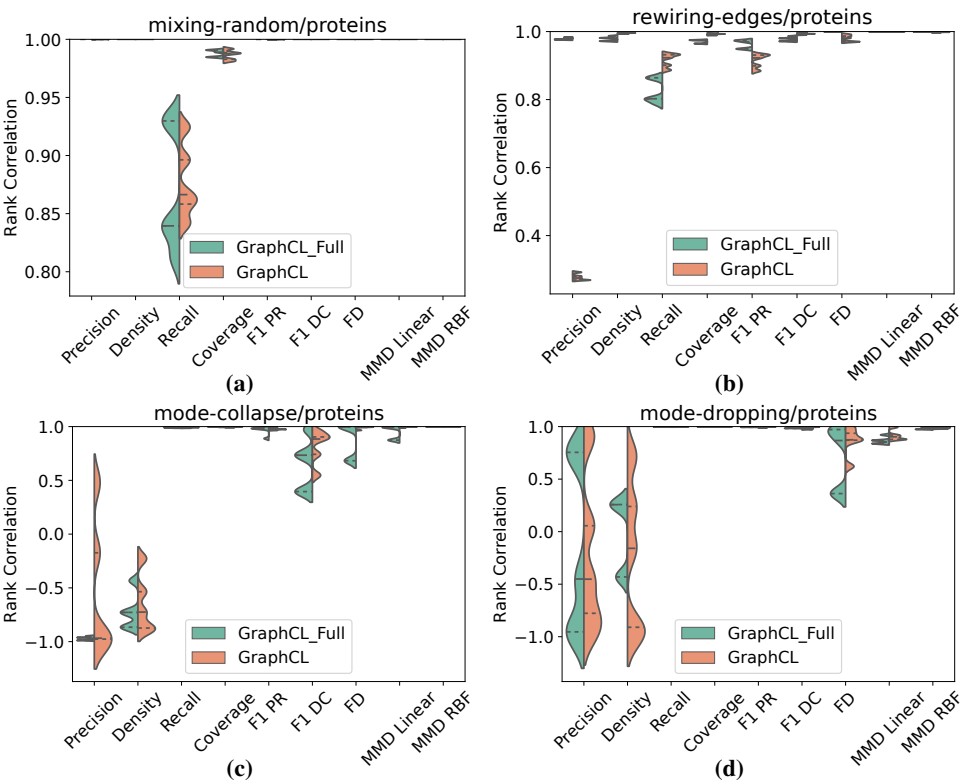

**Figure 18:** Violin comparative results among the methods, with clustering structural features for proteins dataset.

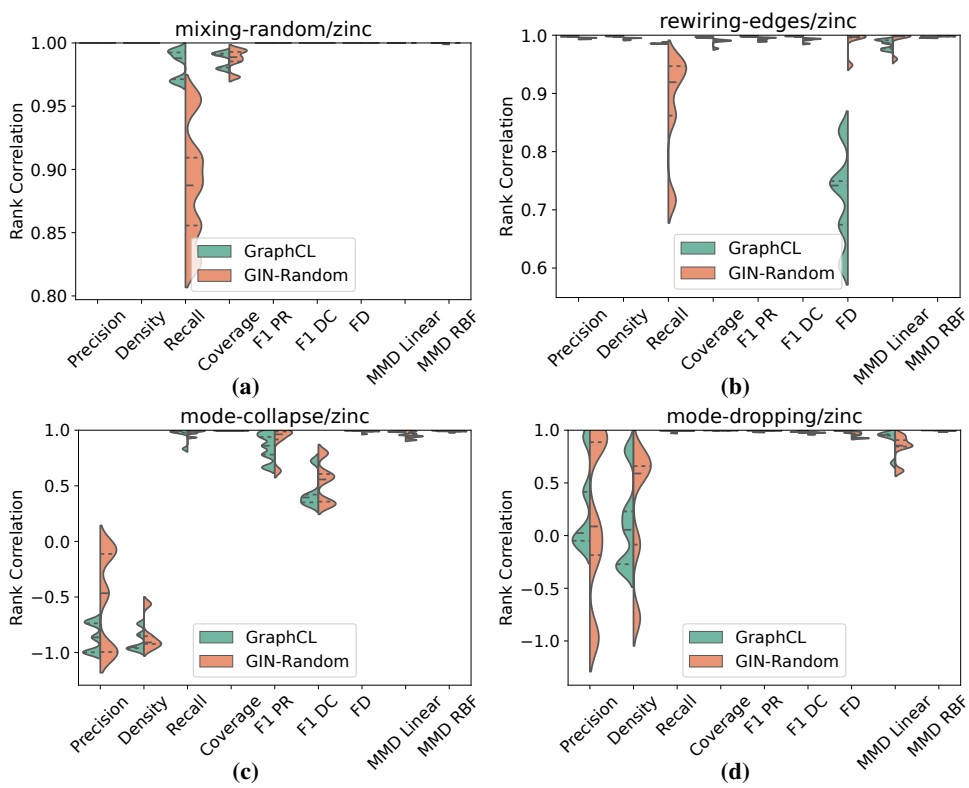

**Figure 19:** Violin comparative results among the methods, with no structural features for zinc dataset.

**Table 10:** Comparison of the main models' mean and median experiment results on the Zinc dataset.

| Experiment | Model Name | Precision | Density | Recall | Coverage | F1PR | F1DC | FD | MMD Lin | MMD RBF |
|---|---|---|---|---|---|---|---|---|---|---|
| Mixing Random | GIN-Random | **1.0/1.0** | **1.0/1.0** | 0.89/0.89 | **0.99/0.99** | **1.0/1.0** | **1.0/1.0** | **1.0/1.0** | **1.0/1.0** | **1.0/1.0** |
| | GraphCL | **1.0/1.0** | **1.0/1.0** | 0.98/0.99 | **0.99/0.99** | **1.0/1.0** | **1.0/1.0** | **1.0/1.0** | **1.0/1.0** | **1.0/1.0** |
| | GraphCL Full | **1.0/1.0** | **1.0/1.0** | **1.0/1.0** | **0.99/0.99** | **1.0/1.0** | **1.0/1.0** | **1.0/1.0** | **1.0/1.0** | **1.0/1.0** |
| Rewiring Edges | GIN-Random | 0.99/0.99 | 0.99/0.99 | 0.88/0.92 | 0.99/0.99 | 0.99/0.99 | 0.99/0.99 | 0.99/**1.0** | 0.99/**1.0** | **1.0/1.0** |
| | GraphCL | **1.0/1.0** | **1.0/1.0** | **0.99/0.99** | **1.0/1.0** | **1.0/1.0** | **1.0/1.0** | 0.72/0.74 | 0.98/0.99 | **1.0/1.0** |
| | GraphCL Full | 0.99/0.99 | 0.99/0.99 | 0.81/0.82 | 0.99/0.99 | 0.99/0.99 | 0.99/0.99 | **1.0/1.0** | **1.0/1.0** | **1.0/1.0** |
| Mode Collapse | GIN-Random | -0.52/-0.47 | -0.84/-0.91 | **0.98/0.99** | **1.0/1.0** | **0.9/0.96** | **0.53/0.56** | 0.98/0.99 | 0.95/0.95 | 0.99/0.99 |
| | GraphCL | -0.86/-0.86 | -0.89/-0.95 | 0.96/0.98 | **1.0/1.0** | 0.85/0.86 | 0.44/0.4 | **1.0/1.0** | **0.98/0.99** | **1.0/1.0** |
| | GraphCL Full | -0.99/-1.0 | -0.86/-0.97 | **0.98/0.99** | **1.0/1.0** | 0.85/0.87 | 0.3/0.34 | **1.0/1.0** | **0.98**/0.98 | **1.0/1.0** |
| Mode Dropping | GIN-Random | 0.16/**0.09** | **0.23/0.59** | **1.0/1.0** | **1.0/1.0** | **1.0/1.0** | 0.97/**0.98** | 0.95/0.96 | 0.84/0.85 | 0.99/0.99 |
| | GraphCL | **0.25**/0.02 | 0.1/0.05 | 0.99/**1.0** | **1.0/1.0** | 0.99/0.99 | **0.99/0.98** | 0.99/**1.0** | 0.9/**0.95** | **1.0/1.0** |
| | GraphCL Full | -0.3/-0.37 | 0.06/-0.13 | **1.0/1.0** | **1.0/1.0** | 0.99/**1.0** | 0.96/0.96 | 0.99/**1.0** | 0.95/0.95 | **1.0/1.0** |

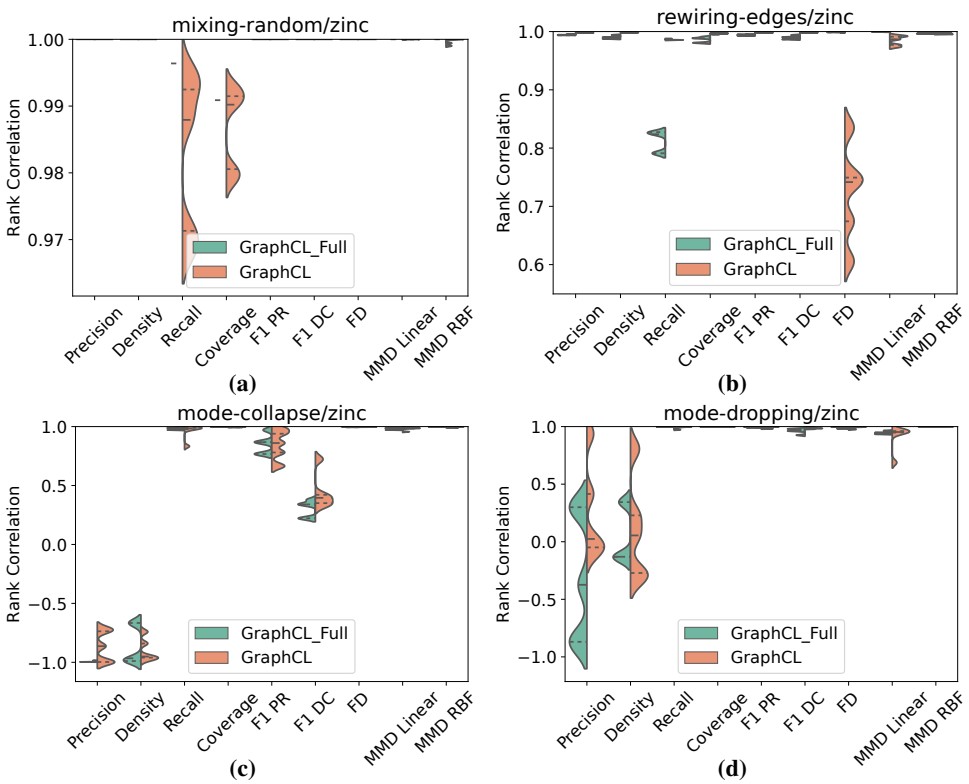

**Figure 20:** Violin comparative results among the methods, with clustering structural features for zinc dataset.