# OpenReview forum: "Evaluating Graph Generative Models with Contrastively Learned Features"
_NeurIPS.cc/2022/Conference — NeurIPS 2022 Accept_

### Official Review · Reviewer_KHXf · 2022-06-28

**Rating:** 6
**Confidence:** 5
**Soundness:** 3 good
**Presentation:** 4 excellent
**Contribution:** 3 good

**Summary:**

The authors propose a new method to evaluate graph generative models by evaluating the representations of the graph which are learned from a contrastive learning approach. They use the training data (which the generative model would also use to train on) and contrastive learning to learn a graph encoder, which they then use to generate representations of the test set and the generated graphs (or in the case of the this paper, of perturbed graphs, which they use to show the effectiveness of the method for evaluation). They show that neither GNNs nor local metrics are strictly better than the other for evaluating graph generative models, since each has a failure case that the other approach can distinguish, and use this as the motivation for trying to combine the respective strengths by using the Graph Substructure Network (which incorporates some local metrics) as their GNN and then train the encoder using contrastive learning and some predefined augmentations. They then assess their proposed method via various graph perturbations, and evaluate the spearman rank correlation between the perturbation and various metrics of interest.

**Questions:**

My questions/suggestions are described in the section above; I look forward to a discussion with the authors.

**Limitations:**

While there is no dedicated limitations section, the authors were upfront about the current limitations (no clear guidelines on how to choose the augmentations at present).

**Strengths And Weaknesses:**

Strengths:
1. The paper is very well written, clear, and easy to follow.
2. The topic is timely and seeks to solve an important problem in the graph representation learning community, namely how to evaluate graph generative models
3. Very nice motivation for why existing approaches (local metrics, random GNNs) are insufficient
4. Using contrastive learning to learn meaningful representations seems like a logical and promising approach to address this issue

Weaknesses:

1. I found it a bit difficult to assess the performance of the approach from the current paper. The visualizations are perhaps too overloaded, which obscured the takeaways (and there are no tables in the main paper).  I think the point is that GraphCL is better than GIN-Random, but the current visualizations don’t give me that immediate impression, they look quite similar. (When plots have so many variables to compare upon, I think it only works to easily digest if there is a clear, dramatic pattern - more subtle differences, even if consistently better, can get lost)  I think the authors should consider alternative visualizations/tables to convey their results more clearly; at a minimum, they could separate out GraphCL and GIN-Random into small multiples of the same plot (but there are probably better solutions).
2. I think the experiments section would also benefit from a bit of restructuring. A lot of space is given to showing the results of not using structural features, using node degree features, and then using node clustering coefficients. I would naturally expect performance to improve as one keeps adding information, so I think there is too much space devoted to this conclusion at the expense of other data/visualization/textual explanations. I think the results you’re trying to convey are 1) graphCL outperforms random-GIN (in whatever comparison format makes most logical sense - e.g. with/without node degree features), and 2) GraphCL with node degree and clustering coefficients (i.e. local measures) further improve performance. I think the experiments section would be stronger if it was organized a bit more around these conclusions and providing the necessary data/plots to make that (or any other conclusions you’d like to highlight) very clear. If there are other conclusions from your experiments that I missed please feel free to include those as well.
3. In general, the plots would benefit from a bit more explaining. For instance, I believe each violin plot contains 6 datasets x 10 random seeds, but it took me a few times searching in the paper to confirm that for myself. I’d make this very explicit, and also make the captions more self-contained.
4. The authors highlight the limitations of using local metrics and using random GNNs, but only compare to using random GNNs. I think the local metrics should be added in as a comparison to confirm that the performance is also better than local metrics.
5. It seems to me that a key benefit of this approach is what the authors stated about graphs being so heterogenous and therefore static approaches (e.g. degree histograms) may appropriately capture the relevant aspects of the data. Contrastive learning has the potential to address this issues since it can use augmentations that are relevant to the dataset, but as the authors themselves note, it is not clear how to choose such subgraphs, and they relegate this choice to future work (where domain specific choice may apply). The choice of augmentations strikes me as a key question in determining how to train the encoder to learn a meaningful representation. I understand this is a hard question to answer, and would be okay accepting this to future work, assuming that the experimental results were strong given the current generic augmentations. But with the current results I am not yet convinced; but am open to being persuaded if the results/visualizations can be more effectively conveyed.
5. (Minor) The graph contrastive learning section in related work reads like a list - I would try to weave this into a more cohesive picture (as is done very nicely in the graph generative model related work section)
6. (Minor) Is Nei(v) neighborhood in Eq. 1? Please define it somewhere.

On the whole, I enjoyed reading this paper, and I think it can be a very compelling paper once the experiments are strengthened and clarified a bit.

---

> ### Author Response · Authors · 2022-08-02
> **Author Response**
>
> Thank you for your feedback. In particular, for the extensive comments about presentation, we’ve begun incorporating it to, we think, improve the presentation of the paper in the current (partial) revision; we’ll continue integrating your suggestions for the final version (continuing to refine the experimental section, rewriting the graph contrastive learning related work section, and so on). We already made some of the changes, which can be seen in the current version. Particularly, we moved degree feature experiments to the appendix and made the other two figures less overcrowded, and added table 1 for comparing mean/median values. We’ll now address some specific questions:
>
>
> > Q: The authors highlight the limitations of using local metrics and using random GNNs, but only compare them to using random GNNs. I think the local metrics should be added in as a comparison to confirm that the performance is also better than local metrics.
>
> We didn’t previously do this because [47] already showed random GINs outperform local metrics on the same evaluation metrics we use (and we outperform random GINs, so we also outperform local metrics). Even so, it might be helpful to see the level of “final improvement” of our methods to local metrics, and so we will add this comparison to the appendix in the final version.
>
>
> > Q: The choice of augmentations strikes me as a key question in determining how to train the encoder to learn a meaningful representation. I understand this is a hard question to answer, and would be okay accepting this for future work, assuming that the experimental results were strong given the current generic augmentations. But with the current results I am not yet convinced; but am open to being persuaded if the results/visualizations can be more effectively conveyed.
>
> We agree that the choice of augmentations is important for contrastive learning over graphs. The graph representation learning community has converged to a set of standard augmentations in recent works, which have been shown to consistently work well across heterogeneous graph datasets. Others, like graph diffusions [19], are also available. For tuning the parameters of these augmentations, while methods like GraphCL rely on trial-and-error, a few recent approaches have demonstrated this can also be done automatically and on-the-fly. For example, LG2AR [20] learns augmentations that can learn to choose the subgraphs to fit the best for the contrastive learning on the dataset at hand. We didn’t use these methods due to their complexity and computational expense; we wanted to show that simple and efficient contrastive methods (e.g., without bi-level optimization [55], policy learning [20] or graph diffusions [19]) still show promising results compared to all previous methods. Indeed, each of our encoders was trained in under 10 minutes on a MacBook Air (using CPU), and still yields good performance.

---

> > ### Comment · Reviewer_KHXf · 2022-08-08
> > **Thank you for your response**
> >
> > I would like to thank the authors for their replies. The answers addressed my concerns from points 4-7. The plots in Figure 3-4 are a bit easier to read now that there are only comparisons between two models, though I still find it difficult to assess performance; because of the difference in range of values among the different metrics many of the metrics / long tails of the distributions many results are so squished at the top that I still find it difficult to conclude from these plots that GraphCL is indeed better. I would encourage the authors to continue to iterate on the plots to find a format that more easily conveys the conclusion they are trying to make. I also appreciate the addition of Table 1, which helps a bit to answer this question, though please also add a reference to Table 1 in the text and bold the best results in the table for easier readability. I also would encourage the authors to continue working on the structure of the experiments section. I saw that the section "Contrastive Training with Structural Information of Node Clustering coefficients" was moved to the appendix, but found that the section was otherwise quite similar to the original submission and correspondingly still found that the conclusions of the experiments (or alternatively framed, the purpose) are not that clear.

---

> > > ### Author Response · Authors · 2022-08-09
> > > **Thanks for your response**
> > >
> > > Thank you for your response and for elaborating on the issues. To address these problems, we have made some changes to the previous version as well as the new version just submitted. However, please note that the rebuttal period is limited. It is our intention to continue working on a final version that will address the problems mentioned. Listed below are some changes and plans for the future:
> > >
> > > 1. In response to your request, we have changed the general structure of the experiments. There are two comparisons we are making:
> > >    a. Random GIN versus GraphCL pertaining
> > >    b. Pretraining with/without structural features
> > > In a final version, we will elaborate more on the purpose, analysis, and conclusions of these experiments. In this regard, having an extra page for a final version will give us more flexibility. We have moved other results to the appendix and will work on their representations for a final version.
> > > 2. The plots in figures 3 and 4 have been improved. The results have been improved slightly plus the distributions have been better represented. As we work toward a final version, we will improve the representations even further.
> > > 3. We made sure to make clear results gathered over all datasets. It is explained in lines 284 and 295, in the explanations of both experiments. In the captions of figures 3 and 4, we also mentioned this.
> > > 4. The captions of figures 3 and 4 are now more self-contained. In the captions, there are further explanations about the experiments.
> > > 5. The best results are now bolded in table 1. Although the table only shows means and medians, it is sufficient to show the improvements our models are making.
> > > 6. We are aware that the appendices continue to follow the previous format. However, these improvements require time and effort, which cannot be accomplished quickly. For the final version, we will make sure that the representations and formats have been improved.

---

### Official Review · Reviewer_Mgpw · 2022-07-09

**Rating:** 6
**Confidence:** 3
**Soundness:** 3 good
**Presentation:** 3 good
**Contribution:** 3 good

**Summary:**

This work mainly argue that we should use contrastively learned representations to evaluate the performance of graph generation methods. The key idea is to use the same training set to train the generative model and the encoder. The encoder is trained by contrastive self-supervised learning and will be used to evaluate the performance of the generative model. Experiments on multiple datasets in terms of multiple metrics are conducted to support the claim.

**Questions:**

(1) Does it take the same resource to train the GIN-random and GraphCL/InfoGraph?

(2) For the encoder, no matter it is GIN-random or GraphCL/InfoGraph, how can we determine if it is trained well? Do we have a validation set for the encoder?

**Limitations:**

Not applicable.

**Strengths And Weaknesses:**

#### Pros ####
(1) The intuition to use contrastively learned representations to evaluate the performance of graph generation methods is sound to me. Basically, contrastive learning teaches model which graphs are similar, thus being consistent with what we expect for generative models. In other words, we also hope that our generative models could generate graphs similar to the training graphs.

(2) The experiment design is reasonable and can support the main claim clearly.

(3) The presentation of this paper is great and easy to follow.

#### Cons ####
(1) The work claims that we should use contrastively learned representations to evaluate graph generative models. However, it does not consider evaluating any generative models in the experiments. It mainly evaluates the correlation between perturbation ratio and the defined value of measurements. Although it can support the claim to a large degree, it could be more convincing if any generative models can be evaluated and analyzed using the proposed strategy.

---

> ### Author Response · Authors · 2022-08-02
> **Author Response**
>
> Thank you for your feedback, which we will use to improve the paper in further revision.
>
> > The work claims that we should use contrastively learned representations to evaluate graph generative models. However, it does not consider evaluating any generative models in the experiments. It mainly evaluates the correlation between perturbation ratio and the defined value of measurements. Although it can support the claim to a large degree, it could be more convincing if any generative models can be evaluated and analyzed using the proposed strategy.
>
> We added a comparison between our method and other evaluation methods for the GRAN [26] generative model over the course of training, in Appendix D.1. We will add the same comparisons for GraphRNN [54] and GraphVAE [24] in the final version.
>
> > Does it take the same resource to train the GIN-random and GraphCL/InfoGraph?
>
> GIN-random doesn’t require any training, as the weights of the network are initialized randomly. GraphCL/InfoGraph require one-time training for each dataset. After training once, the model can be used for all experiments on that dataset. It's especially noteworthy to mention that GraphCL/InfoGraph are very efficient methods: for the experiments in our paper, each encoder took under 10 minutes to train on CPU using a MacBook Air.
>
> > For the encoder, no matter if it is GIN-random or GraphCL/InfoGraph, how can we determine if it is trained well? Do we have a validation set for the encoder?
>
> We rely on the contrastive loss over the validation set to determine if the model is trained.

---

> > ### Comment · Reviewer_Mgpw · 2022-08-02
> > **Thanks**
> >
> > Thanks for the clarification.

---

> > > ### Author Response · Authors · 2022-08-02
> > > **Thank you**
> > >
> > > Thank you. Please do let us know if you had any further questions or requests for clarification.

---

### Official Review · Reviewer_tjmf · 2022-07-11

**Rating:** 6
**Confidence:** 3
**Soundness:** 2 fair
**Presentation:** 3 good
**Contribution:** 2 fair

**Summary:**

The paper proposes a self-supervised learning based approach to generate representations of real and synthetic graphs  in order to evaluate the quality of latter.  The general idea is to train graph encoders on the same data used to train a graph generator, using self-contrastive learning and then use these trained encoders to embed generated graphs and compare the embeddings with the same of real graphs in a test set. The claim is that such contrastively trained GNNs are more capable of extracting meaningful features from the graphs, compared to randomly initialized GNNs. The choice of perturbation schemes used in contrastive learning to train graph encoders could be domain specific. To demonstrate the merit of the idea, the paper chooses Graph Isomorphism Network (GIN) as the encoder architecture and ideas from two well-known graph self-supervised learning works (GraphCL and InfoGraph) to pretrain the encoders. The encoders are pre-trained on six datasets (three synthetic and three real-world) and then used to encode the real graphs as well as synthetic graphs generated from the four traditional graph-perturbation algorithms: Mixing-Random, Rewiring Edges, Mode Collapse, and Mode Dropping. To evaluate the goodness of proposed encoders, authors monitor the correlation (higher the better) between the degree of graph-perturbations and eight discrepancy metrics (commonly used to compare similarity between two graph distributions) between the distributions of latent representations for real graphs and generated graphs.  The proposed approach is compared against state-of-the-art technique where GIN is initialized randomly and the results indicate better correlation values for proposed approach as opposed  to SOTA approach. Additionally, authors also show that graph encoders supplemented with features capturing graph substructural information further aids to the distinguishing power of graph encoders.

**Questions:**

1. Just to confirm, is there a particular reason to choose the contrastive learning schemes from InfoGraph and GraphCL in this paper?

2. All the results in Figures 3-6 correspond to which of the six datasets? Or are the results combined across all the datasets?

3. In several plots, I observe that rank correlations seem to be better for GIN-Random than the proposed approach (e.g. Figure 4(b), F1DC, F1PR, Coverage metrics). Am I reading the graph correctly? And if so, any explanation for why that might be the case?



**Limitations:**

I will recommend authors to talk about the limitations in the context of weaknesses I highlighted above if they agree with it.

**Strengths And Weaknesses:**

Strengths:
1. The paper highlights the limitations of using generic local metrics to evaluate the two distributions.
2. The idea of using graph-substructural features in training graph encoders along with Lipschitz control is quite plausible and also shown to work well.

Weaknesses:
1. The technical contribution of the paper is somewhat limited. The performance of the proposed evaluator is largely dependent on the choice of perturbation scheme used in contrastive learning as well as the architecture of the encoder. The right choices for each of the two components could largely vary with application domains and thus it may not be obvious in every application. And in case it is obvious, wouldn't that pretty much seal the choice of graph generator?

2. It might be difficult to use the proposed approach to evaluate two graph generators G1 and G2 that use different perturbation schemes P1 and P2 respectively. In such a case, what perturbation scheme shall be used to train the evaluation encoder?

3. For certain discrepancy metrics, the SOTA evaluator seem to be performing better than the proposed evaluator (see comments below), but it is not explained well as to why this might be happening.

---

> ### Author Response · Authors · 2022-08-02
> **Authors Response**
>
> Thanks for your comments, which will help us to improve the paper in revision. We address some questions and concerns below.
>
> > Q: The technical contribution of the paper is somewhat limited. The performance of the proposed evaluator is largely dependent on the choice of perturbation scheme used in contrastive learning as well as the architecture of the encoder. The right choices for each of the two components could largely vary with application domains and thus it may not be obvious in every application. And in case it is obvious, wouldn't that pretty much seal the choice of graph generator?
>
> We think this paper tackles a very important problem for graph generative modeling, demonstrating for the first time that representation learning can improve the quality of evaluation metrics for graph generative models. We are also introducing principled techniques fusing high-order structural information and Lipschitz smoothness to enhance the quality of evaluations even further.
>
> We’d like to emphasize (and will edit to make this clearer in the paper) the difference between the related notions of “augmentation” and “perturbation.” In the terminology we use, we are using augmentations (node dropping, edge perturbation, attribute masking, and sub-graph “cropping”) to train an encoder to learn graph representations. On the other hand, perturbations (mixing-random, rewiring edges, mode collapse, and mode dropping) are used as an evaluation tool to compare our method to others. We adopted these perturbations based on the choices of the previous SOTA method [47].
>
> For the choice of augmentations, the graph representation learning community has converged to a set of standard augmentations in recent works, which have been shown to consistently work well across heterogeneous graph datasets [20, 55, 56, 58]. It is also shown that the frequency of sampling these augmentations can be automatically tuned only based on training data, so they can be adapted to various applications on-the-fly. Other augmentations are of course available, such as graph diffusion [19], but these four augmentations are shown to be robust and efficient to compute.
>
> For the choice of encoder, we chose GIN because: (1) it is shown to be an expressive GNN, and (2) previous works used it to evaluate their performance, so using the same type of encoder seems helpful for a fair comparison. This is analogous to how in the computer vision community, the vast majority of generative model evaluations use an InceptionV3 [44] model trained on ImageNet, since it is shown to be strong enough and a common baseline is helpful.
>
> Even so, picking another set of base encoders (e.g. GCN, GAT, etc.) or another set of graph augmentations (whether something generic like graph diffusion or something domain-specific) is completely independent of the choice of graph generator: these should be chosen based on representation learning performance, not tuned for the down-stream task (i.e. getting our preferred generative model to look the best). As we discuss, it is possible to choose more domain-specific augmentations to tailor to a particular setting, but it does not seem to be necessary, since these generic augmentations work well: we view this as a significant strength of our approach, which does not _require_ but does easily _allow_ domain-specific adaptation. It is easy to imagine settings where we know that certain augmentations are likely to be helpful without implying a particular generative model is then the right thing: for instance, incorporating knowledge of chemistry to molecular graphs, although physical chemistry is far too complex for us to immediately know a “correct” generative model.
>
>
>
> > Q: It might be difficult to use the proposed approach to evaluate two graph generators G1 and G2 that use different perturbation schemes P1 and P2 respectively. In such a case, what perturbation scheme shall be used to train the evaluation encoder?
>
> As we just discussed, the choice of perturbation is unrelated to actually evaluating graph generators, but is simply a technique for telling whether our metrics behave as we would like them to behave.
>
> It seems, though, that by P1, P2 you may instead be referring to what we call the choice of augmentations used in training a model. This is also independent of the choice of encoder: our evaluation method is totally independent of the generation scheme, and treats each generator simply as a black-box set of samples.

---

> > ### Author Response · Authors · 2022-08-02
> > **Author Response**
> >
> > > Q: For certain discrepancy metrics, the SOTA evaluator seem to be performing better than the proposed evaluator (see comments below), but it is not explained well as to why this might be happening.
> >
> > Answered below.
> >
> >
> > > Q: Just to confirm, is there a particular reason to choose the contrastive learning schemes from InfoGraph and GraphCL in this paper?
> >
> > These two contrastive approaches show good performance across different datasets from different domains and are fairly efficient to train. Much previous work provides node-level representations rather than the graph-level representations we require (e.g., [45, 50, 57, 58]); we also did not consider some more computationally expensive methods (e.g., [19, 20, 55]). There is nothing in our approach particularly coupled to InfoGraph or GraphCL, however; one can plug in any preferred representation learning method for the encoder.
> >
> >
> > > Q: All the results in Figures 3-6 correspond to which of the six datasets? Or are the results combined across all the datasets?
> >
> > As mentioned in line 263, these plots combine results from all datasets: each point corresponds to a single random seed on a single dataset (we’ve clarified this further in the figure captions in our current revision). The corresponding figures for individual datasets are in Appendix D.
> >
> >
> > > Q: In several plots, I observe that rank correlations seem to be better for GIN-Random than the proposed approach (e.g. Figure 4(b), F1DC, F1PR, Coverage metrics). Am I reading the graph correctly? And if so, any explanation for why that might be the case?
> >
> > The plots here, known as violin plots, show the full distribution of results. For the cases you flagged, note in particular that our median performance (the white dot inside the violin plot in the submitted version) is substantially better than the median performance of GIN-Random. In these instances, however, our worst individual run (the bottom of the “violin”) was indeed worse than the worst run of GIN-Random. In the current version, this plot has moved to the appendix D.1. As discussed around line 282, these poor cases are for the small Lobster and Grid datasets.
> >
> > Since submission, we’ve seen that with additional regularization, our method works better on these datasets, probably due to “overfitting” less. Our revised draft shows these improvements.
> >
> > To try to avoid this confusion, we’ve also changed the presentation format in our current revision, to a table showing medians and means as well as separate (more legible) violin plots showing the distribution.

---

> > > ### Comment · Reviewer_tjmf · 2022-08-08
> > > **Thanks**
> > >
> > > Thanks to the authors for elaborate responses and clarifications. I will consider them while revising my rating.

---

### Author Response · Authors · 2022-08-07
**Paper Discussion**

Dear Reviewers,

We tried our best to address your questions and comments, and we have been eagerly waiting to address any further questions that you had. As the rebuttal period is coming to an end, we would like to ask you to please engage with us so we can further improve the quality of the paper.

Thank you.

---

### Meta-Review · Area_Chair_2oNM · 2022-08-26

**Recommendation:** Accept
**Confidence:** Certain

**Metareview:**

This paper proposed a new way of evaluating the quality of graph generative models, by leveraging the contrastively learned representations. The reviewers generally found the study is interesting and important, and the idea of leveraging contrastive learning for evaluation is also sound and novel. I also believe the work would be impactful for future research in the graph generative model domain, and I highly encourage the authors to properly release easy to use benchmark tools/datasets around it, if possible.


**Award:**

No

---

### Decision · Program_Chairs · 2022-09-14

Accept